# Antigenic mapping and functional characterization of human New World hantavirus neutralizing antibodies

Taylor B Engdahl[1†], Elad Binshtein[2†], Rebecca L Brocato[3†], Natalia A Kuzmina[4,5], Lucia M Principe[3], Steven A Kwilas[3], Robert K Kim[3], Nathaniel S Chapman[1], Monique S Porter[1], Pablo Guardado-Calvo[6], Félix A Rey[6], Laura S Handal[2], Summer M Diaz[2], Irene A Zagol-Ikapitte[7], Minh H Tran[7], W Hayes McDonald[7], Jens Meiler[8], Joseph X Reidy[2], Andrew Trivette[2], Alexander Bukreyev[4,5,9], Jay W Hooper[3*], James E Crowe[1,2,10*]

[1]Department of Pathology, Microbiology and Immunology, Vanderbilt University, Nashville, United States; [2]Vanderbilt Vaccine Center, Vanderbilt University Medical Center, Nashville, United States; [3]Virology Division, United States Army Medical Research Institute of Infectious Diseases, Ft Detrick, United States; [4]Department of Pathology, The University of Texas Medical Branch at Galveston, Galveston, United States; [5]Galveston National Laboratory, Galveston, United States; [6]Institut Pasteur, Université Paris Cité, Paris, France; [7]Department of Biochemistry and Mass Spectrometry Research Center, Vanderbilt University, Nashville, United States; [8]Department of Chemistry, Vanderbilt University, Nashville, United States; [9]Department of Microbiology and Immunology, University of Texas Medical Branch, Galveston, United States; [10]Department of Pediatrics, Vanderbilt University Medical Center, Nashville, United States

*For correspondence:
Jay.w.hooper.civ@health.mil
(JWH);
james.crowe@vanderbilt.edu
(JEC)

[†]These authors contributed equally to this work

**Abstract** Hantaviruses are high-priority emerging pathogens carried by rodents and transmitted to humans by aerosolized excreta or, in rare cases, person-to-person contact. While infections in humans are relatively rare, mortality rates range from 1 to 40% depending on the hantavirus species. There are currently no FDA-approved vaccines or therapeutics for hantaviruses, and the only treatment for infection is supportive care for respiratory or kidney failure. Additionally, the human humoral immune response to hantavirus infection is incompletely understood, especially the location of major antigenic sites on the viral glycoproteins and conserved neutralizing epitopes. Here, we report antigenic mapping and functional characterization for four neutralizing hantavirus antibodies. The broadly neutralizing antibody SNV-53 targets an interface between Gn/Gc, neutralizes through fusion inhibition and cross-protects against the Old World hantavirus species Hantaan virus when administered pre- or post-exposure. Another broad antibody, SNV-24, also neutralizes through fusion inhibition but targets domain I of Gc and demonstrates weak neutralizing activity to authentic hantaviruses. ANDV-specific, neutralizing antibodies (ANDV-5 and ANDV-34) neutralize through attachment blocking and protect against hantavirus cardiopulmonary syndrome (HCPS) in animals but target two different antigenic faces on the head domain of Gn. Determining the antigenic sites for neutralizing antibodies will contribute to further therapeutic development for hantavirus-related diseases and inform the design of new broadly protective hantavirus vaccines.

## Editor's evaluation

Antibodies perform a critical function in host defense against viruses and have emerged as major therapeutic tools in modern medicine, as evidenced by the large-scale use of antibody-based therapies during the COVID-19 pandemic. This paper describes the characterization of human antibodies to hantaviruses that have the potential to create devastating epidemics. The results teach us about the viral structures that are targets for neutralization and the results are relevant for vaccine development and antibody therapeutic design. The evidence provided is convincing and the results are important and should be of interest to immunologists, virologists, and those working on antibody engineering and therapeutic antibodies.

## Introduction

Hantaviruses are emerging zoonotic pathogens that are endemic worldwide and classified into two categories based on the geographic distribution of their reservoir hosts and pathogenesis in humans (*Jonsson et al., 2010*). Almost 60 virus species have been identified in rodents or shrews, and over 20 species cause disease in humans (*Laenen et al., 2019*). Old World hantaviruses, including Hantaan virus (HTNV), Puumala virus (PUUV), Dobrava -Belgrade virus (DOBV), and Seoul virus (SEOV), mainly occur in Eastern Europe and China and cause hemorrhagic fever with renal syndrome (HFRS). New World hantaviruses (NWHs), including Sin Nombre virus (SNV) and Andes virus (ANDV), are endemic in North and South America and cause hantavirus cardiopulmonary syndrome (HCPS). Person-to-person transmission of ANDV has been reported, including in a recent outbreak in Argentina resulting in 34 confirmed cases and 11 fatalities (*Martinez et al., 2005*; *Martínez et al., 2020*). There are no current FDA-approved medical countermeasures to prevent or treat hantavirus-related disease.

The viral glycoproteins, designated Gn and Gc, form a hetero-tetrameric spike on the surface of the hantavirus virion and facilitate attachment and entry. Previous studies have reported crystal structures for the Gn (*Li et al., 2016*; *Rissanen et al., 2017*) and Gc (*Guardado-Calvo et al., 2016*; *Willensky et al., 2016*) ectodomains of PUUV and HTNV, and recent work has described the molecular organization of Gn/Gc on the virion surface (*Serris et al., 2020*). Gc is a class II fusion protein and undergoes conformational changes triggered by low pH to mediate the fusion of the viral and host endosomal membranes. Gn is proposed to play a role in receptor attachment and stabilize and prevent the premature fusogenic triggering of Gc (*Mittler et al., 2019*; *Bignon et al., 2019*). Structural studies have identified a capping loop on Gn that shields the fusion loop on Gc, and the glycoprotein complex is thought to undergo dynamic rearrangements between a closed (or capped) and open (or uncapped) form of the spike (*Serris et al., 2020*; *Bignon et al., 2019*).

Recent efforts have identified features of the molecular basis of neutralization by some antibodies targeting HTNV Gn (*Rissanen et al., 2021*) or PUUV Gc (*Rissanen et al., 2020*). A rabbit-derived antibody, HTN-Gn1, targets domain A on Gn and overlaps with the putative binding sites for other murine-derived HTNV (*Arikawa et al., 1992*) and ANDV mAbs (*Duehr et al., 2020*). An antibody isolated from a bank vole, P-4G2, targets a site spanning domain I and II on Gc that is occluded in the post-fusion trimeric form, suggesting that the antibody may neutralize through blocking conformational changes required for fusion (*Willensky et al., 2016*; *Rissanen et al., 2020*). Although mAb P-4G2 neutralizes PUUV and ANDV, this activity has only been tested in pseudovirus neutralization assays, and it is unknown if the antigenic site on Gc is occluded on the surface of the authentic virus. Also, these antibodies were not derived from human B cells and were not induced by natural infection, but rather were elicited following inoculation of animals immunization with recombinant protein, recombinant VSV constructs, or virus-infected tissues. It is unclear what sites are accessible to antibodies during a natural infection, and if human antibodies target antigenic sites that differ from those recognized by rodents. The first clues toward the sites of vulnerability on the Gn/Gc spike for the human antibody response were recently described by *Mittler et al., 2022*. A panel of 135 antibodies were isolated from convalescent PUUV donors, and two distinct neutralizing sites were determined by negative stain electron microscopy (nsEM); one at the Gn/Gc interface and one on prefusion exposed surface of Gc domain I. ADI-42898, a quaternary-site mAb, demonstrated cross-clade neutralizing activity and protected in both PUUV bank vole and ANDV hamster post-exposure challenge models. However, mAbs targeting Gn were not described, and, due to its level of surface exposure, the N-terminal domain of Gn likely represents a major site of the neutralizing human antibody response (*Serris*

*et al., 2020*; *Engdahl and Crowe, 2020*). Gn exhibits a higher degree of sequence variability than Gc, likely indicating that Gn is under more immune pressure than Gc (*Li et al., 2016*).

Previously, we characterized a panel of human mAbs isolated against Gn/Gc from survivors of SNV or ANDV infection (*Engdahl et al., 2021*). We demonstrated that NWHs antibodies target at least eight distinct sites on the ANDV Gn/Gc complex, four of which contained potently neutralizing antibody clones, but the location of those sites on the Gn/Gc spike was not known. Here, we define four distinct neutralizing antigenic sites on the Gn/Gc complex. Two potently neutralizing species-specific antibodies, ANDV-5 and ANDV-34, map to two non-overlapping epitopes in the Gn ectodomain and neutralize the virus by blocking attachment. We also show that these antibodies are similar to two human antibody clones previously isolated, MIB22 and JL16 (*Garrido et al., 2018*). Broadly neutralizing antibodies, ANDV-44 and SNV-53, map to the interface of Gn/Gc, while a less potently neutralizing but broadly reactive antibody, SNV-24, targets domain I of Gc. Both classes of broad antibodies function by inhibiting the triggering of fusion. These data shed light on the basis for both ANDV-specific and broad hantavirus recognition by these antibody clones and suggest that hantavirus vaccine designs should focus on effectively eliciting antibodies to these sites of vulnerability.

## Results

### Neutralizing antibodies target at least four sites on ANDV GnH/Gc spike

The hantavirus glycoprotein spike forms a complex hetero-tetrameric structure on the surface of the virus and is composed of two proteins, Gn and Gc (*Guardado-Calvo and Rey, 2021a*). Gn is separated into two main domains: an N-terminal, membrane-distal Gn head domain (GnH), and a C-terminal Gn base domain (GnB) (*Li et al., 2016*; *Serris et al., 2020*). GnH likely functions in viral attachment, but also has a 'capping loop' region that interfaces with the Gc fusion loop to prevent premature exposure of the hydrophobic loop (*Bignon et al., 2019*). Due to the surface exposure and sequence variability of GnH, it is likely immunodominant compared to Gc and GnB[17]. GnB promotes the tetramerization of the complex and is shielded from immune recognition in the Gn/Gc complex. We previously isolated a panel of 36 mAbs from human survivors who were naturally infected with SNV or ANDV, and we demonstrated that the mAbs target eight sites on Gn/Gc, several of which are sites of vulnerability for neutralization. We selected five representative mAbs from this panel for further study to determine the

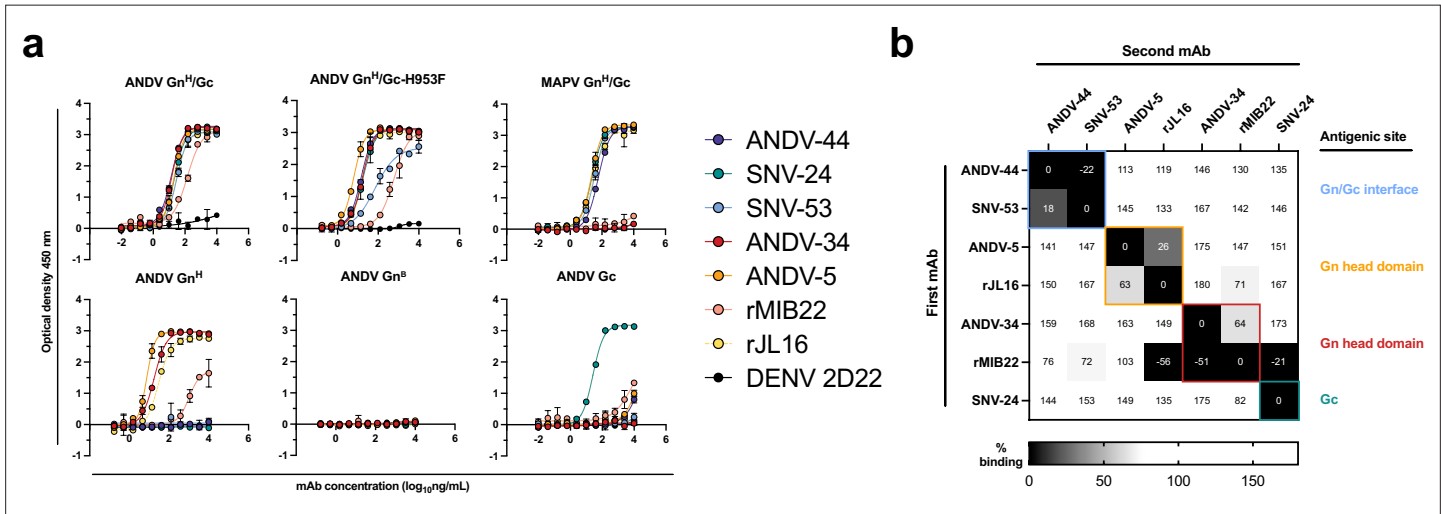

**Figure 1.** Hantavirus neutralizing antibodies target four distinct regions on the glycoprotein spike. (**a**) Binding potency of mAbs to recombinant hantavirus antigens, ANDV GnH/Gc, ANDV GnH/Gc_H953F, MAPV GnH/Gc, ANDV GnH, ANDV GnB, ANDV Gc, expressed in S2 cells. Binding curves were obtained using non-linear fit analysis, with the bottom of curve constrained to 0, using Prism software. The data shown are representative curves from 3 independent experiments. Mean ± SD of technical duplicates from one experiment are shown. (**b**) Competition binding analysis of neutralizing antibodies to ANDV GnH/Gc recombinant protein measured using BLI. % Competition is designated by the heatmap, where black boxes indicate complete competition, gray boxes indicate intermediate competition, and white boxes indicate no competition. The data are shown are representative from two independent experiments.

features of the neutralizing antigenic sites. We also produced IgG1 forms of two previously reported human mAbs based on publicly available cDNA sequences, MIB22 or JL16 (*Garrido et al., 2018*), and cloned them into a human IgG1 expression vector for recombinant expression (hereafter we designate these IgGs as rMIB22 or rJL16).

To determine which subunits of the spike were targeted by neutralizing antibodies, we expressed and purified hantavirus antigens in *Drosophila* S2 cells and tested for antibody binding reactivity to the $Gn^H$, $Gn^B$, or Gc monomeric proteins or the $Gn^H$/Gc heterodimer (*Figure 1a*). We also generated and purified a 'stabilized' form of $Gn^H$/Gc by introducing a H953F mutation that prevents the Gc protein from making conformational changes to the post-fusion form (*Serris et al., 2020*). All the mAbs tested demonstrated reactivity to the linked ANDV $Gn^H$/Gc construct by ELISA, as well as to the 'pre-fusion' stabilized form (ANDV $Gn^H$/Gc_H953F). Most of the mAbs, apart from ANDV-34 and rMIB22, demonstrated reactivity to Maporal (MAPV) $Gn^H$/Gc, a closely related species to ANDV. ANDV nAbs (ANDV-34, ANDV-5, rMIB22, and rJL16) all displayed binding reactivity to $Gn^H$, with $EC_{50}$ values less than 100 ng/mL except for rMIB22 ($EC_{50}$ value 5.6 µg/mL). SNV-24 was the only antibody that displayed reactivity to Gc, and we did not detect binding reactivity to $Gn^B$ for any mAbs tested. Notably, bnAbs ANDV-44 and SNV-53 did not have detectable binding to $Gn^H$, $Gn^B$, or Gc alone, and only bound to the linked $Gn^H$/Gc antigen, suggesting these antibodies bind a quaternary site only present on the Gn/Gc heterodimer.

We also performed competition-binding studies utilizing biolayer interferometry and tested the seven mAbs for binding to the linked $Gn^H$/Gc antigen (*Figure 1b*). As previously shown with a cell surface-displayed version of Gn/Gc, bnAbs ANDV-44 and SNV-53 bin to similar competition groups while SNV-24 bins to a distinct site on Gc. We also determined that rJL16 bins in a group with ANDV-5, while rMIB22 bins with ANDV-34. However, rMIB22 and rJL16 also asymmetrically compete for binding, indicating that all four mAbs likely bind in close spatial proximity to each other on the $Gn^H$ domain.

## Low likelihood of viral escape for neutralizing hantavirus antibodies

Determining escape mutations is helpful for designing vaccines and therapeutics to treat hantavirus-related diseases. We next sought to identify neutralization-resistant viral variants for each of the seven nAbs described in this study. To do this, we employed two different methods: (1) a high-throughput escape mutant generation assay using real-time cellular analysis based on similar assays previously described (*Gilchuk et al., 2020*; *Greaney et al., 2021*; *Suryadevara et al., 2022*) and (2) serial passaging of virus in increasing concentrations of neutralizing antibodies (*Figure 2a*). Previously, we demonstrated that our mAbs showed similar neutralization potencies for VSV-pseudotyped viruses and authentic viruses (*Engdahl et al., 2021*). The one class of mAbs that demonstrated a notable discrepancy was that of Gc-targeting mAbs, such as SNV-24, which showed an ~1000 fold lower neutralization potency for authentic viruses compared to VSV-pseudotyped viruses. Based on the general similarities between the two systems, we considered the VSV surrogate system appropriate to study escape mutation generation of escape mutant glycoprotein sequences. Thus, for all neutralization assays, we employed pseudotyped VSVs bearing the glycoproteins Gn/Gc from either the SNV or ANDV species. For RTCA-based escape mutant generation, we tested each antibody at saturating neutralization conditions and evaluated escape based on delayed cytopathic effect (CPE) in each individual replicate well. Thus, if a replicate well demonstrated a delayed drop in cellular impedance, this finding indicated the presence of an escape mutant virus and was noted out of the total number of replicate wells to give a percentage of escape. Notably, we did not detect escape mutants using this method for most neutralizing antibodies (e.g. SNV-53, ANDV-44, SNV-24, ANDV-34, or ANDV-5; *Figure 2a*). For example, we could not detect any escape mutants through this method for SNV-53, even though we attempted 368 replicates with the VSV/SNV virus and 184 replicates with the VSV/ANDV virus for selection (*Figure 2a*).

In contrast, a CPE profile for resistance was noted for 92% or 100% of the replicates for rMIB22 or rJL16, respectively (*Figure 2a*). We confirmed resistance to either rMIB22 or rJL16 at a 10 µg/mL concentration of the selecting mAb and sequenced the M-segment gene of the escaped virus from six replicates. All sequenced rMIB22-selected viruses bore the mutation K76T. Due to the high number of replicates demonstrating a neutralization escape phenotype and the fact that all the escape viruses isolated had the K76T mutation, it is likely that this escape mutation was present at a high proportion in the viral preparation's original stock. Five rJL16-selected virus sequences had an L224R mutation,

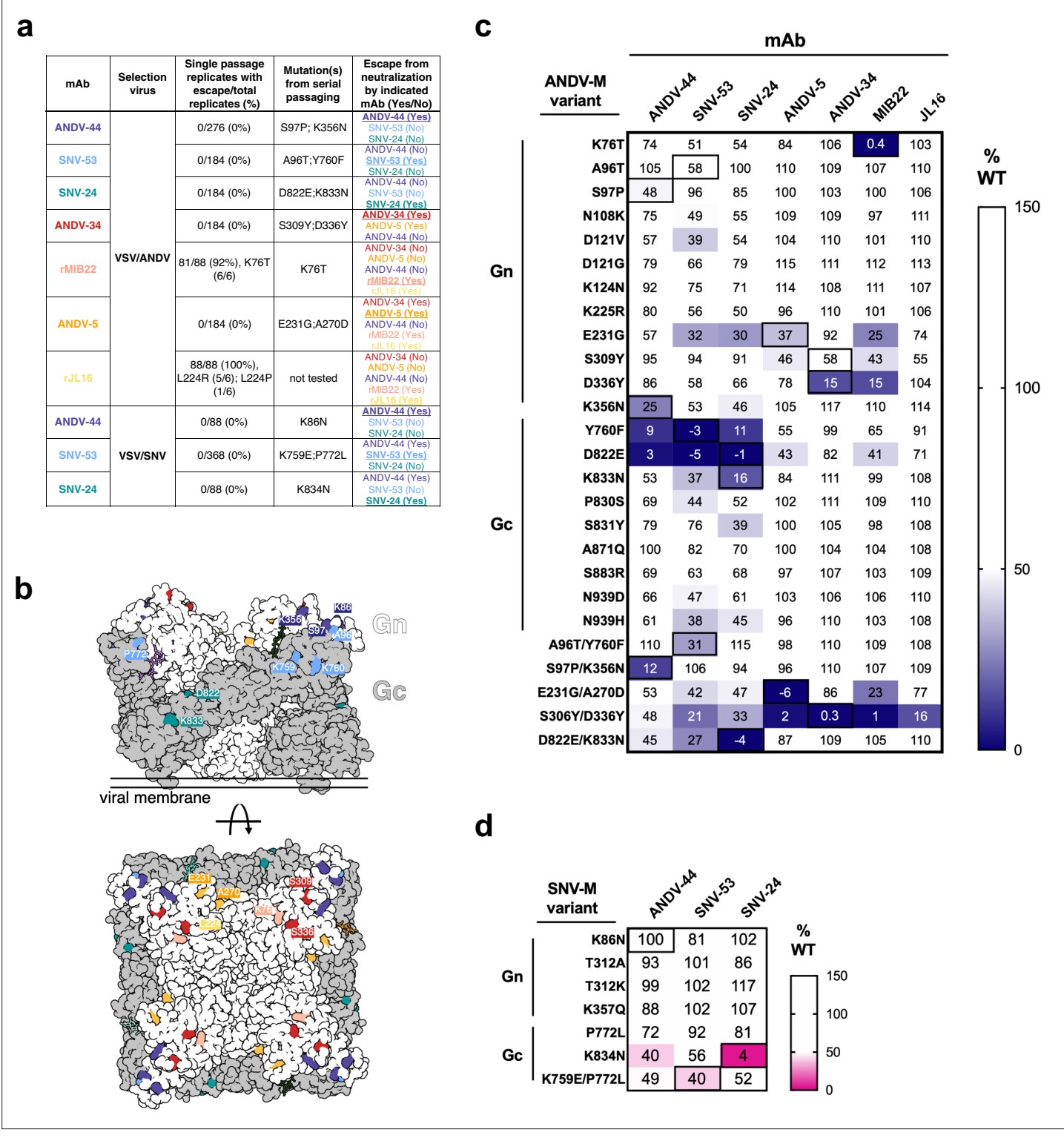

**Figure 2.** Escape mutant generation and mutagenesis mapping indicate critical binding residues for hantavirus mAbs. (**a**) Results from viral escape selection for indicated antibodies. Real-time cellular analysis escape mutant mapping shows the number of replicates with escape over the total number of replicates for each selection mAb against the indicated selection virus. Mutations from serial passaging were identified for each mAb, and escape was confirmed in the presence of saturating mAb concentrations. VSV/ANDV or VSV/SNV were used for escape selections. (**b**) Side and top view of escape mutants mapped to the ANDV Gn/Gc spike (PDB: 6ZJM). The colored spheres designate escape mutants for the indicated antibody. Gn is shown in white, and Gc is shown in grey. (**c**) Heatmap of mAb binding in the presence of ANDV mutant constructs. Dark blue boxes indicate loss of binding. The black boxes designate escape mutants for the indicated antibody. The percent binding (% WT) of each mAb to the mutant constructs was compared to

*Figure 2 continued on next page*

*Figure 2 continued*

the WT SNV or ANDV control. The data are shown as average values from three to four independent experiments. All numbering for ANDV sequences was based on GenBank AF291703.2 and SNV sequences were based on GenBank KF537002.1. (**d**) Heatmap of SNV mutant constructs as described in **c**. All numbering for SNV sequences were based on GenBank KF537002.1.

The online version of this article includes the following source data and figure supplement(s) for figure 2:

**Source data 1.** Percent mAb binding in the presence of ANDV mutants.

**Source data 2.** Percent mAb binding the presence of SNV mutants.

**Figure supplement 1.** Mutagenesis expression levels and gating strategy.

**Figure supplement 2.** Amino acid alignment of hantavirus species.

and one had an L224P mutation. Overall, these results indicate that identifying viral escape from hantavirus neutralizing antibodies using this method is possible but suggests a low likelihood of in vitro escape for most of the potently neutralizing hantavirus nAbs targeting multiple distinct antigenic regions we selected for study.

## Mapping mutations in antibody escape variant viruses selected with ANDV-specific nAbs or bnAbs

Although we could not select escape mutants in a high-throughput, single passage approach for five of the mAbs, we still wanted to map the critical binding residues and identify escape mutations for the neutralizing antibodies of interest. Thus, we identified escape mutants for mAbs through serial passaging in cell culture monolayers in the presence of antibody and confirmed neutralization-resistant phenotypes for each escape mutant virus (*Figure 2—figure supplement 1*).

Potent species-specific neutralizing antibodies selected for mutations located in Gn$^H$. ANDV-34-selected viruses contained mutations S309Y and D336Y, two residues on the surface exposed face of Gn domain B (*Figure 2b*). rMIB22-resistant variant viruses contained a single mutation, K76T, located in domain A of Gn in close spatial proximity to S336, corresponding with binning analysis for these mAbs (*Figure 2b*). Neutralization-resistant viruses selected by ANDV-5 contained mutations E231G and A270D, mapping to α–3 and α–4 helices of the Gn head domain, respectively. Both residues are located near L224, identified in rJL16-resistant viruses, supporting previous data that ANDV-5 and rJL16 compete for a similar binding site (*Figure 1b*).

For bnAbs SNV-53 and ANDV-44, we identified mutations in the interface region between Gn/Gc (*Figure 2a*). Variant viruses selected by ANDV-44 or SNV-53 contained mutations in both the Gn ectodomain (K86 on the VSV/SNV background and K356, S97, A96 on the VSV/ANDV background) and Gc domain II near the highly conserved fusion loop (K759, P772 on the VSV/SNV background and Y760 on the VSV/ANDV background). For SNV-24, we selected escape mutants D822E and K833N in the variant VSV/ANDV viruses and a single homologous mutation, K834N, in the escape-resistant VSV/SNV. Residues K86, Y760, D822, and K833 are highly conserved among members of the *Orthohantavirus* genus (*Figure 2—figure supplement 2*).

## Mapping of ANDV-specific nAb and bnAb epitopes by mutagenesis

Since most of the neutralization-resistant viruses contained multiple mutations, we next sought to identify which mutant residues impacted antibody binding. We generated a panel of single- or double-point mutants in the SNV or ANDV M-segment genes based on escape mutants that we selected with mAbs and previously published VSV/ANDV escape mutants (*Duehr et al., 2020*; *Garrido et al., 2018*). We expressed each mutant M segment gene on the surface of Expi293F cells and used a flow cytometric binding assay to assess how each variant impacted mAb binding. The % wild-type (WT) values were generated by dividing the binding of the mutant by that of the WT construct, and values were normalized based on a positive control oligoclonal mix of mAbs.

The expression levels of SNV and ANDV M-segment variants were comparable to that of the WT constructs, except for three single-point mutants: A270D, C1129F, and K759E (*Figure 2—figure supplement 1b*). All three mutations were identified in infectious VSVs, so it is not clear why we could not detect the expression of the mutated proteins. Notably, these residues all co-occurred with additional mutations, indicating that these residues may be found in functionally constrained sites.

Most single-point mutants we identified partially impacted the corresponding mAb binding reactivity (*Figure 2c and d*). However, the loss-of-binding phenotype was most evident for the double-mutant M-segment genes, suggesting that multiple mutations are required to generate neutralization-resistant variants. We identified a few single mutations that impacted the binding of multiple broadly-neutralizing mAbs. For example, Y760F (a residue near the fusion loop) ablates the binding of ANDV-44/SNV-53 (Gn/Gc interface) and SNV-24 (Gc domain I). Another single point-mutant, D822E (located in Gc domain I), also reduced the binding of all bnAbs. This finding may indicate that this altered residue promotes rearrangements of Gn/Gc that have allosteric effects on interface mAb binding. Additionally, one double mutant, S309Y/D336Y, exhibited a moderate reduction of the binding of the bnAbs and lost binding of ANDV-specific nAbs tested. Both residues are located on Gn domain B and could represent a critical evolutionary strategy to escape species-specific immunity. If amino acid changes in highly immunogenic epitopes on Gn domain B do not impact viral fitness, then genetic changes in these sites may occur over time or in the case of a large outbreak.

## BnAbs recognize two antigenic sites on Gn/Gc

Next, we sought to determine the location of the antigenic sites targeted by bnAbs, SNV-53 and SNV-24, using negative-stain electron microscopy of MAPV Gn$^H$/Gc in complex with antigen-binding fragments (Fabs) of antibodies (*Figure 3a*). To identify and dock the right Fabs to the maps, we used the information from the binding groups (*Figure 1*) and escape mutations (*Figure 2*). The 3D reconstructions (EMD-26735) of SNV-53 and SNV-24 in complex with the MAPV Gn$^H$/Gc heterodimer showed that the Fabs localize to two distinct regions in proximity to the corresponding viral escape mutations selected (*Figure 3b*). SNV-24 bound to Gc domain I, an epitope that is only accessible in the pre-fusion state of Gc and is near the interface between inter-spike Gc heterodimers. P-4G2 (a bank vole-derived mAb) (*Rissanen et al., 2021*) and group II human PUUV mAbs described previously (*Mittler et al., 2022*) also map to a similar site on Gc. This site has limited accessibility in the full virus lattice structure due to neighboring interspike contacts and may results in the incomplete neutralizing activity to authentic virus we demonstrated by these mAbs previously (*Engdahl et al., 2021*). Based on escape mutants generated and nsEM reconstructions, SNV-53 may interact with the capping loop region on Gn, which interfaces with the *cd* and *bc* loops on domain II of Gc (*Figure 3b and c*; *Serris et al., 2020*). This 3D reconstruction supports our previous finding that bnAbs SNV-53 and ANDV-44 target a quaternary epitope only accessible on the Gn/Gc heterodimer and is consistent with the broadly protective site targeted by ADI-42898 (*Mittler et al., 2022*). Modeling SNV-53 Fab in the context of the (Gn-Gc)$_4$ lattice structure shows possible clashes with adjacent spikes in the full virus assembly (*Figure 3—figure supplement 1*). However, the low resolution of the model may not fully recapitulate the binding angle and we previously demonstrated that this mAb was capable of complete neutralization of authentic viruses. We also performed hydrogen-deuterium exchange mass spectrometry (HDX-MS) with the Fab forms of SNV-53 or ANDV-44 in complex with ANDV Gn$^H$/Gc. In the capping loop region, a reduction in deuterium uptake was seen for peptides of ANDV-44 (peptide spanning amino acids 80–100 and peptide spanning 82–100) while the deuterium uptake for SNV-53 remained unchanged. (*Figure 3—figure supplement 2*).

## Structural basis for neutralization by Gn-targeting antibodies

We were also interested in the epitopes targeted by ANDV-specific mAbs elicited during natural infection, thus, we performed 3D reconstructions of ANDV-5 and ANDV-34 in complex with the ANDV Gn$^H$ protomer (EMD-26736), which showed that the mAbs bind to ANDV Gn$^H$ at two distinct sites in concordance with the results from competition binning (*Figure 1b*). Fab ANDV-5 bound the Gn head domain α3 – α4 angled parallel to the membrane and may interact with a neighbor Gn protomer in the (Gn-Gc)$_4$ complex (*Figure 4a*). Fab ANDV-34 bound to a distinct site on the opposite face of Gn corresponding to domain B and is angled perpendicular to the viral membrane in the (Gn-Gc)$_4$ complex (*Figure 4a*).To understand the molecular level interaction of the ANDV-specific neutralizing antibodies, ANDV-5 and ANDV-34, and Gn$^H$, we collected a cryo-EM data set and reconstructed a 3D map at 4.1 Å resolution (*Figure 4b* and *Figure 4—figure supplement 1*, *Supplementary file 1*). The heavy chain solely drives the interactions between ANDV-5 and Gn with four hydrogen bonds and excessive hydrophobic interactions (*Figure 4c and e*). The CDRH3 loop interacts with a hydrophobic cavity on Gn (*Figure 4—figure supplement 2*). The escape mutation

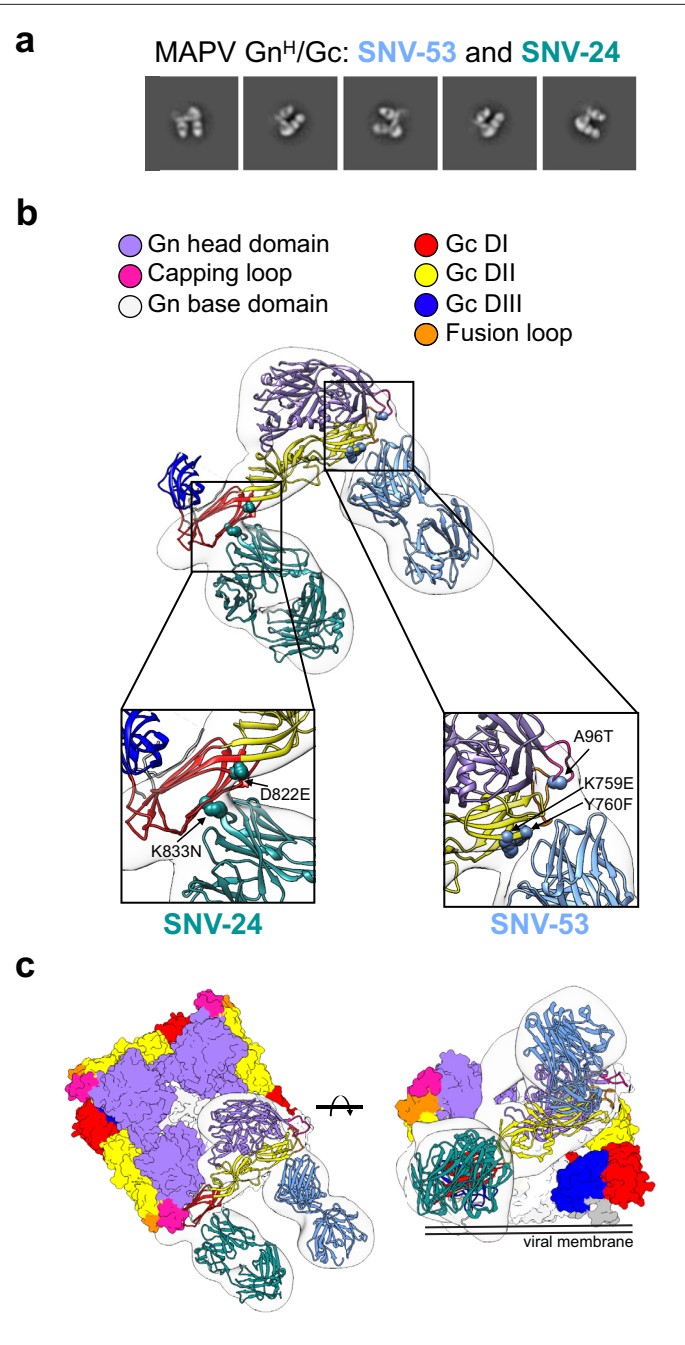

**Figure 3.** BnAbs SNV-53 and SNV-24 target two sites on the Gn^H/Gc heterodimer. (**a**) Representative nsEM 2D-class averages of SNV-53 and SNV-24 Fabs in complex with MAPV Gn^H/Gc heterodimer. (**b**) Surface representations (light grey) of SNV-53 (blue) and SNV-24 (green) in complex with MAPV Gn^H/Gc. Escape mutations are indicated by the colored spheres. Gn^H is colored in purple, Gn^B is colored in light grey, and the capping loop is colored in pink. Domain I, II, and III of Gc are colored in red, yellow, and blue, respectively, and the fusion loop is colored in orange. (**c**) Model of bnAbs in complex with the (Gn-Gc)_4 spike as colored in **b**, top and side view are shown.

The online version of this article includes the following figure supplement(s) for figure 3:

**Figure supplement 1.** SNV-53 Fab docked to the hantavirus surface glycoprotein lattice (EMD-11236).

**Figure supplement 2.** Hydrogen-deuterium exchange mass spectrometry analysis of hantavirus antibodies in complex with ANDV Gn^H/Gc.

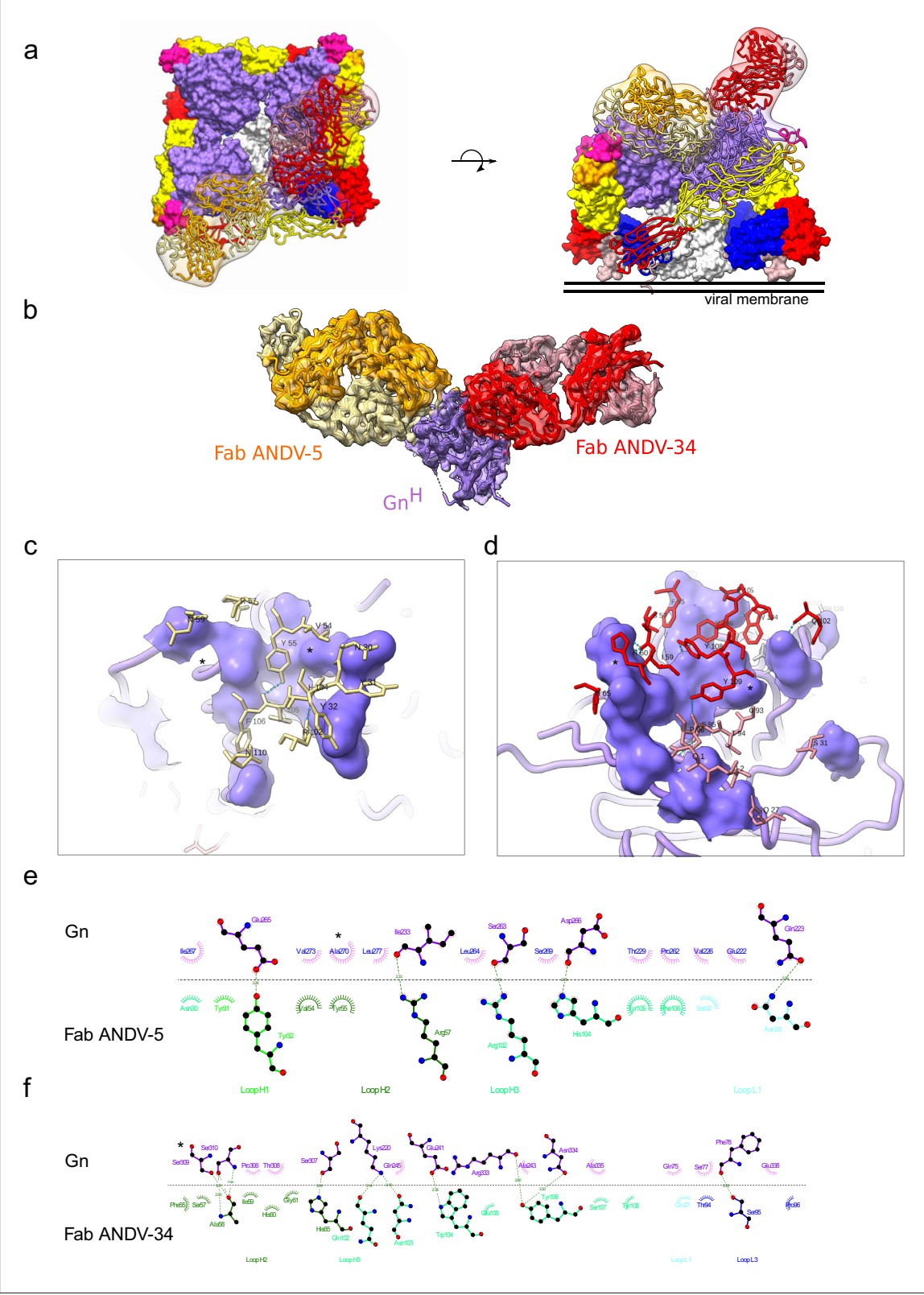

**Figure 4.** Cryo-EM structure of neutralizing antibodies ANDV-5 and ANDV-34 in complex with ANDV Gn^H. (**a**) Top view (*right*) and side view (*left*) of the heterotetramer Gn/Gc (***Serris et al., 2020***) (PDB: 6ZJM) with low resolution map and model of Gn^(H) ANDV-5 and ANDV-34 (purple, orange and red, respectively) complex superimpose with Gn^H. (**b**) Cryo-EM map and model of the Gn-Fabs complex. Right, transparent EM map color by chain with Gn purple, ANDV-5 heavy chain in yellow, ANDV-5 light chain in orange, ANDV-34 heavy chain in red and ANDV-34 light chain in pink. (**c**) Zoom-in on the

*Figure 4 continued on next page*

*Figure 4 continued*

paratope/epitope interface of the ANDV-5:Gn^H. Fab ANDV-5 residues that is in close contact with Gn. Heavy chain residues (yellow stick), label with single letter and residue number. Gn is shown in purple with the contact residues shown in a surface representation. Asterisks correspond to residues found in the escape mutant viruses. Blue dashed line, H-bond. (**d**) Zoom-in on the paratope/epitope interface of the ANDV-34:Gn^H. Fab ANDV-34 residues that is in close contact with Gn. Heavy residues (red stick), light chain residues (pink stick), label with single letter and residue number. Gn is shown in purple with the contact residues shown in a surface representation. Asterisks correspond to residues found in the escape mutant viruses. Blue dashed line, H-bond. (**e**) Fab ANDV-5 paratope and epitope residues involved in hydrogen bonding (dashed lines) and hydrophobic interactions. Hydrophobic interactions residues are shown as curved lines with rays. Atoms shown as circles, with oxygen red, carbon black, and nitrogen blue. Interacting residues that belong to CDR loops are colored in different shade. Image was made with Ligplot+49 (*Laskowski and Swindells, 2011*). (**f**) Fab ANDV-34 paratope and epitope residues involved in hydrogen bonding (dashed lines) and hydrophobic interactions. Hydrophobic interactions residues are shown as curved lines with rays. Atoms shown as circles, with oxygen red, carbon black, and nitrogen blue. Interacting residues that belong to CDR loops are colored in different shade. Image was made with Ligplot+49.

The online version of this article includes the following figure supplement(s) for figure 4:

**Figure supplement 1.** Workflow of cryo-electron microscopy processing for model of ANDV-5 and ANDV-34 Fabs in complex with ANDV Gn^H.

**Figure supplement 2.** Residue interaction plot of Gn^H with ANDV-5 or ANDV-34.

---

Ala270 is located in the center of the epitope. We could not detect expression of the Gn/Gc glycoproteins bearing the A270D mutation, indicating that A270D may solely ablate ANDV-5 binding but may incur fitness costs offset by mutations (i.e. E231G) that are not located in the epitope. The ANDV-34 interface with Gn is more extensive than the ANDV-5:Gn interface and includes both heavy and light chain interactions with 11 hydrogen bonds (*Figure 4d and f*). Both the escape mutations Ser309 and Asp336 are part of the interface, and Ser309 has one hydrogen bond with the epitope.

## Neutralization potency is dependent on bivalent interactions

We have previously demonstrated that all antibody clones described here neutralize ANDV or SNV as full-length IgG1 molecules (*Engdahl et al., 2021*). Here, we sought to determine if bivalency was required for NWH neutralization; therefore, we generated the five antibody clones as recombinant Fab molecules and tested the neutralizing activity to VSV constructs bearing ANDV or SNV glycoproteins. BnAb ANDV-44 retained some neutralizing activity as the Fab form to VSV/ANDV, while SNV-53 Fab form did not have detectable neutralizing activity to VSV/ANDV (*Figure 5a*). The opposite was true against VSV/SNV, with SNV-53 demonstrating a decrease in potency, while the ANDV-44 Fab form had no detectable neutralizing activity to VSV/SNV. Gc-specific bnAb SNV-24 only had a slight decrease in neutralization potency in the Fab form. Although it is important to note that we previously demonstrated that SNV-24 has an ~1000 fold decrease in $IC_{50}$ neutralization values between the VSV/SNV and the authentic SNV; thus, this finding may not be representative of authentic virus neutralization.

Gn-specific mAb ANDV-5 retained neutralizing activity as a Fab but showed a slight decrease in neutralization potency (*Figure 5b*). We did not detect any neutralizing activity of the Fab form of ANDV-34, despite there being no detectable difference in the binding of Gn^H between the IgG1 and Fab form of ANDV-34. This result could indicate that this antibody's neutralization depends on steric hindrance, not avidity effects. As expected, we did not detect neutralizing activity of ANDV-5, ANDV-34, rMIB22, and rJL-16 to VSV/SNV (*Figure 5b*).

## ANDV-specific antibodies neutralize through receptor blocking

New World hantaviruses use PCDH-1 as a receptor for cell attachment (*Jangra et al., 2018*). We previously demonstrated that ANDV-5 IgG competes with the extracellular cadherin 1 domain (EC1) of PCDH-1, the primary domain that engages with the Gn/Gc complex. Here, we show that the Fab form of ANDV-5 also blocks both EC1 and EC1-EC2 binding, suggesting that ANDV-5 binds to the receptor-binding site (RBS), although the residues comprising the RBS are currently unknown (*Figure 5c*). We also tested mAbs rJL16 and rMIB22 for receptor blocking activity, and both antibodies significantly reduced EC1 and EC1-EC2 binding to ANDV Gn/Gc. Although ANDV-34 does not block EC1 or EC1-EC2 binding to Gn/Gc, it competes for binding to Gn/Gc with rMIB22 and thus likely binds near the receptor-binding site.

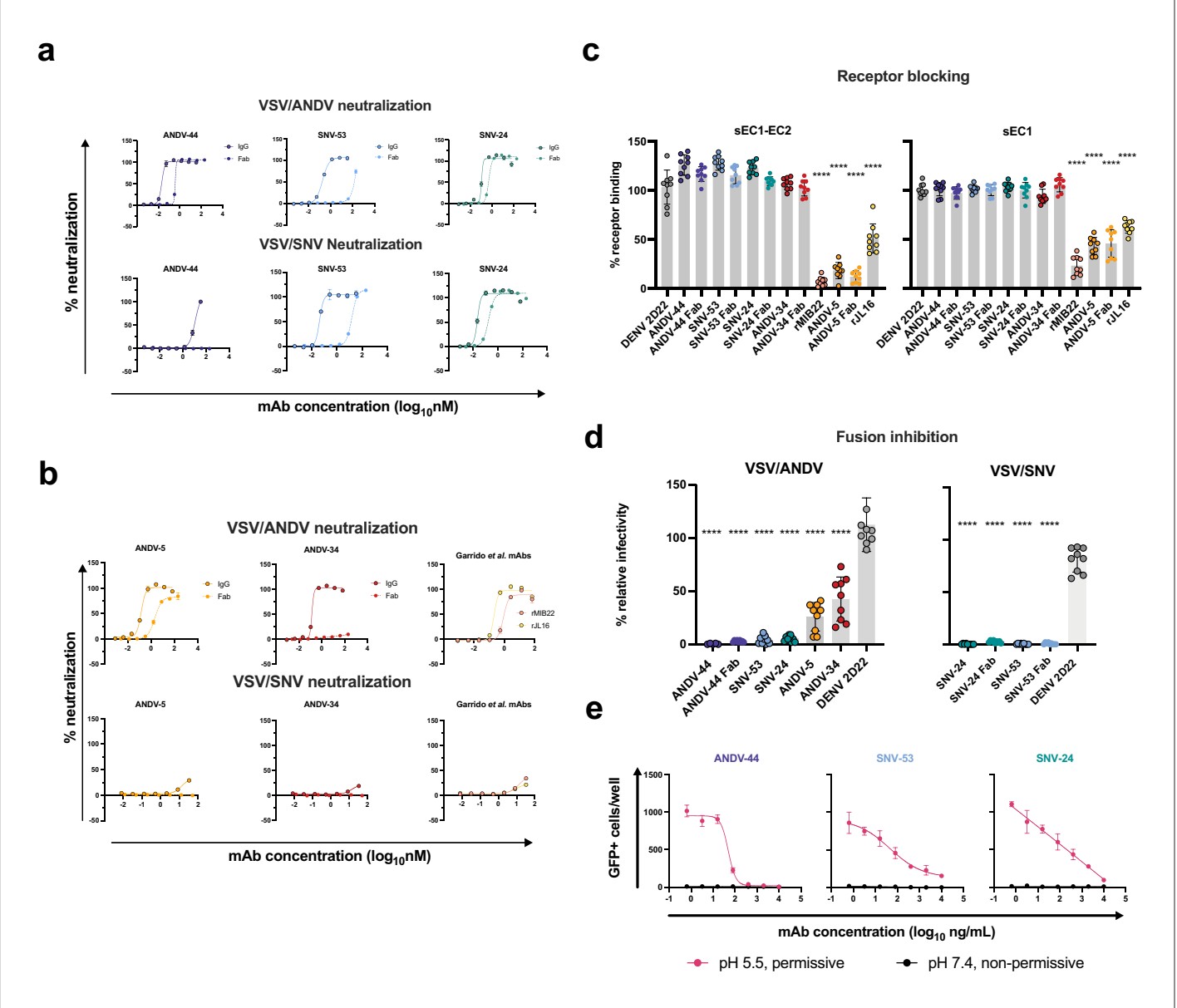

**Figure 5.** Potently neutralizing hantavirus mAbs inhibit viral entry through viral attachment blocking and/or fusion inhibition. (**a**) Neutralization curves of IgG1 and Fab forms of broad mAbs (SNV-53, ANDV-44 or SNV-24) to VSV/SNV and VSV/ANDV determined through real-time cellular analysis. The data shown are representative curves from three independent experiments. Mean ± SD of technical duplicates from one experiment are shown.(**b**) Neutralization curves of IgG1 and Fab forms of ANDV-nAbs (ANDV-5, ANDV-34, rJL16, or rMIB22) to VSV/SNV and VSV/ANDV determined through real-time cellular analysis. The data shown are representative curves from three independent experiments. Mean ± SD of technical duplicates from one experiment are shown. (**c**) sEC1 or sEC1-EC2 blocking activity of neutralizing antibodies determined through a flow cytometric assay, in which mAbs were added at saturating concentration before the addition of the each soluble PCDH-1 domain labeled with Alexa Fluor 647 dye. The data shown are averages ± SD from three experiments, n=9. One-way ANOVA with Dunnett's test, ****p<0.0001; ns, non-significant. (**d**) FFWO assay testing VSV/ANDV or VSV/SNV post-attachment antibody neutralization in a permissive (pH 5.5) conditions at 30 µg/mL. The data shown are averages ± SD from three experiments, n=9. One-way ANOVA with Dunnett's test, ****p<0.0001. (**e**) Representative curves of dose-dependent VSV/ANDV fusion inhibition. The data shown are average values for technical replicates ± SDs. The experiments were performed three times independently with similar results.

## BnAbs neutralize the virus through fusion inhibition

The broadly reactive antibodies did not have detectable receptor-blocking activity in assays with EC1 of PCDH-1, possibly because Old World hantaviruses do not engage with PCDH-1. Thus, to further explore if the bnAbs neutralize virus at a pre- or post-attachment step in the life cycle, we performed

a fusion-from-without (FFWO) assay to test the ability of each antibody to neutralize virus after attachment. All mAbs tested significantly reduced viral infectivity of VSV/ANDV in Vero cell culture monolayers (*Figure 5d and e*). However, ANDV-44 completely neutralized VSV/ANDV in the FFWO assay as a full-length IgG1 or Fab molecule. BnAbs SNV-53 and SNV-24 also reduced VSV/ANDV infectivity post-attachment, but both antibodies fully neutralized VSV/SNV as IgG1 or Fab molecules (*Figure 5d*). SNV-24 binding maps to the fusogenic protein Gc, while SNV-53 also had critical binding residues located in the capping loop on Gn and the fusion loop on Gc. This result suggests that for the clones studied here, bnAbs target highly conserved fusogenic regions of the Gn/Gc complex and function to neutralize the virus principally through fusion inhibition.

## Somatic mutations enable broad recognition

Previous sequence analysis determined that SNV-53, which is encoded by human antibody variable region gene segments *IGHV5-51*01/IGLV1-40*01*, is remarkably close to the germline-encoded sequence, with a 97 or 95% identity to the inferred heavy and light chain variable gene sequences, respectively (*Engdahl et al., 2021*). To understand if somatic mutations are necessary broad recognition of this epitope, we aligned the antibody coding sequence with the inferred germline gene segments and reverted all mutations in the antibody variable regions to the residue encoded by the inferred germline gene. When reverted to its germline form, SNV-53 loses binding activity to all hantavirus species except for SNV (*Figure 6a*). The germline reverted form of ANDV-44 also has decreased binding activity to ANDV and HTNV, although it has similar reactivity to SNV. SNV-53 also loses all detectable neutralizing activity to both VSV/ANDV and VSV/SNV in its germline reverted form, while ANDV-44 shows decreased potency to VSV/ANDV in its germline-encoded form (*Figure 6b*). These findings indicate that breadth is developed through affinity maturation, although the critical amino acid changes important for broad reactivity of this quaternary site are unknown.

## Common gene usage may contribute to ANDV neutralization

ANDV-5 and ANDV-34 are both encoded by *IGVH1-69* v-genes and use F alleles (*IGHV1-69*01* and *\*06*, respectively). Notably, the heavy chain of MIB22 is also encoded by *IGHV1-69*06*, indicating that this gene usage is commonly employed in the neutralizing response found in multiple individuals in response to hantavirus infection. Although all three mAbs are germline-encoded by an F allele at position 55 in the CDHR2 loop, ANDV-34 retains the phenylalanine, but ANDV-5 mutates the residue to a tyrosine, which is a similarly bulky hydrophobic residue, while MIB22 has an F54S mutation (*Figure 6e*). MIB22 retains the conserved isoleucine at position 54, but ANDV-5 and ANDV-34 mutate the isoleucine to valine, a similarly hydrophobic side chain. Notably, Y55 and V54 make hydrophobic interactions with A270 and neighboring residues. The germline reverted ANDV-5 shows substantially reduced binding to ANDV Gn/Gc and no detectable neutralizing activity to VSV/ANDV (*Figure 6c and d*). Thus, somatic hypermutation was critical to the potency of ANDV-5, and it is possible that the F55Y/I54V mutation was critical to the interactions that ANDV-5 makes with Gn. MIB22 and ANDV-34 both compete for binding to Gn; thus, *IGHV1-69*06* may encode important germline characteristics that are used to bind to this antigenic site on Gn. F55 hydrophobic interactions with S309, a residue identified in ANDV-34 neutralization resistant viruses. Interestingly, when ANDV-34 is reverted to its germline form, it has a similar binding and neutralization potency as rMIB22 (*Figure 6d*).

## SNV-53 MAb cross-protects against an Old Word hantavirus when administered prior to exposure

We previously demonstrated that ANDV-44, SNV-53 or SNV-24 provided protection from a lethal ANDV challenge in hamsters (*Engdahl et al., 2021*). Since SNV-53 potently neutralized HTNV in vitro, we next tested if the bnAb could protect hamsters from HTNV infection in an animal prophylaxis model. Syrian golden hamsters are highly susceptible to HTNV infection. Antibodies can be tested in this model for reduction of viral load and prevention of seroconversion after challenge, although the virus does not cause discernable pathological changes in tissues (*Custer et al., 2003*; *Hooper et al., 2001a*; *Perley et al., 2020*). Hamsters were administered 5 mg/kg of SNV-53, SAB-159 (positive control), or DENV 2D22 (negative control) by the i.p. route one day before exposure by the i.m. route with 200 PFU of HTNV (*Figure 7a*). The polyclonal human antibody immune globulins SAB-159 were developed through hyperimmunization of transchromosomic bovines with HTNV, and PUUV DNA

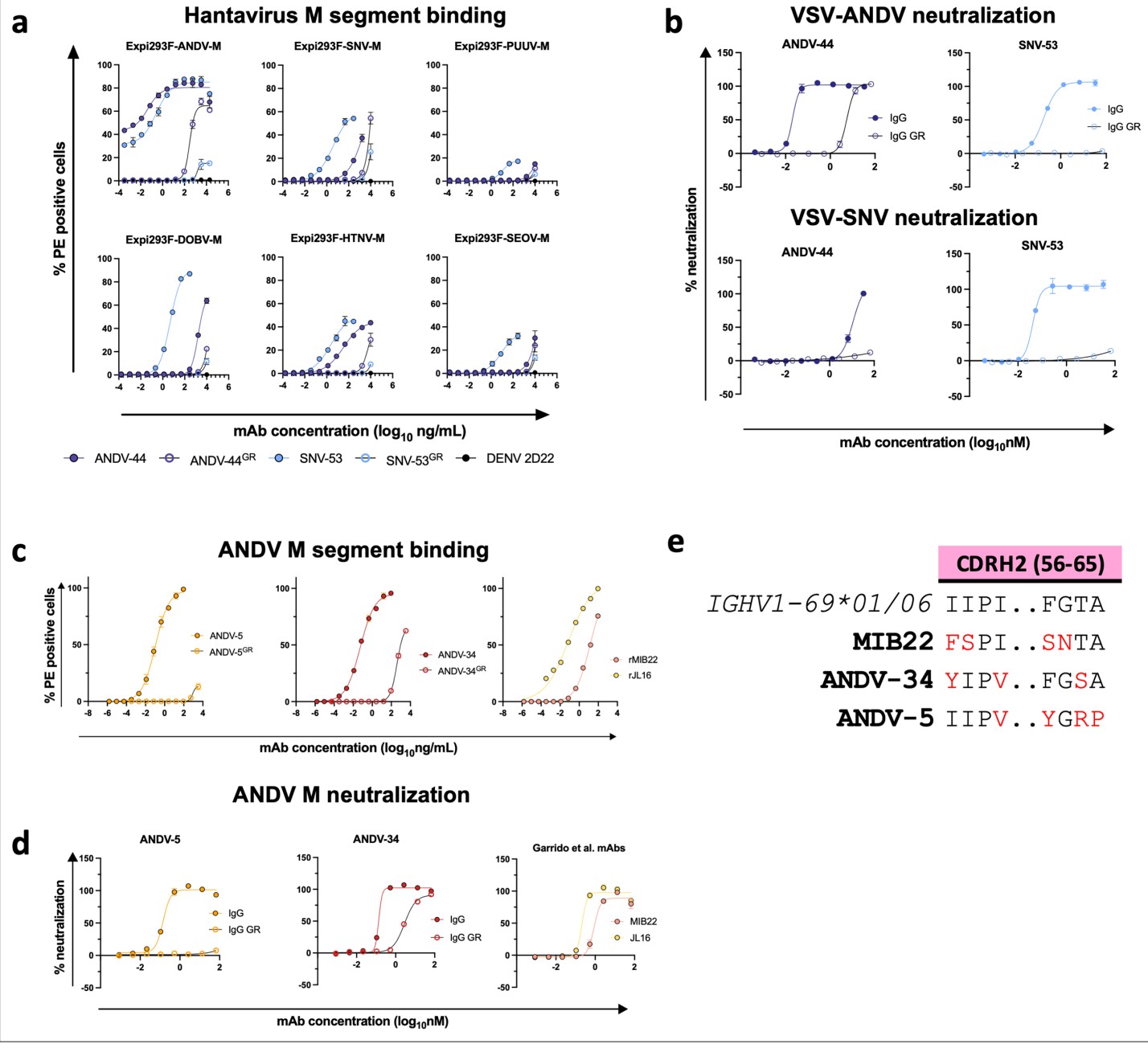

**Figure 6.** Reactivity and potency of germline revertant forms of NWH antibodies. (**a**) Representative binding curves for all germline reverted forms of SNV-53 and ANDV-44 bnAbs to Expi293F cells transfected with ANDV, SNV, PUUV, DOBV, HTNV, or SEOV Gn/Gc. The value for % PE+ cells was determined by gating on cells stained only with secondary antibodies. Data shown are average values for technical replicates ± S.D. The experiment was performed 3 times independently with similar results; one experiment is shown. (**b**) Representative neutralization curves for all bnAbs to VSVs bearing ANDV or SNV glycoproteins Gn/Gc. % Neutralization was measured using real-time cellular analysis and calculated by comparing CPE in the treatment wells with a cells only control well. Data shown are average values for technical replicates ± S.D. The experiment was performed 3 times independently with similar results. (**c**) Representative binding curves for all ANDV-nAbs, including germline reverted forms, to Expi293F cells transfected with ANDV, as described above in **a**. (**d**) Representative neutralization curves for all ANDV-nAbs to VSVs bearing ANDV, as described above in **a**. (**e**) Alignment of MIB22, ANDV-34, and ANDV-5 CDHR2 sequences to the human germline IGHV1-69 gene. Somatically mutated residues are indicated in red. Alignment was generated using IMGT/DomainGapAlign.

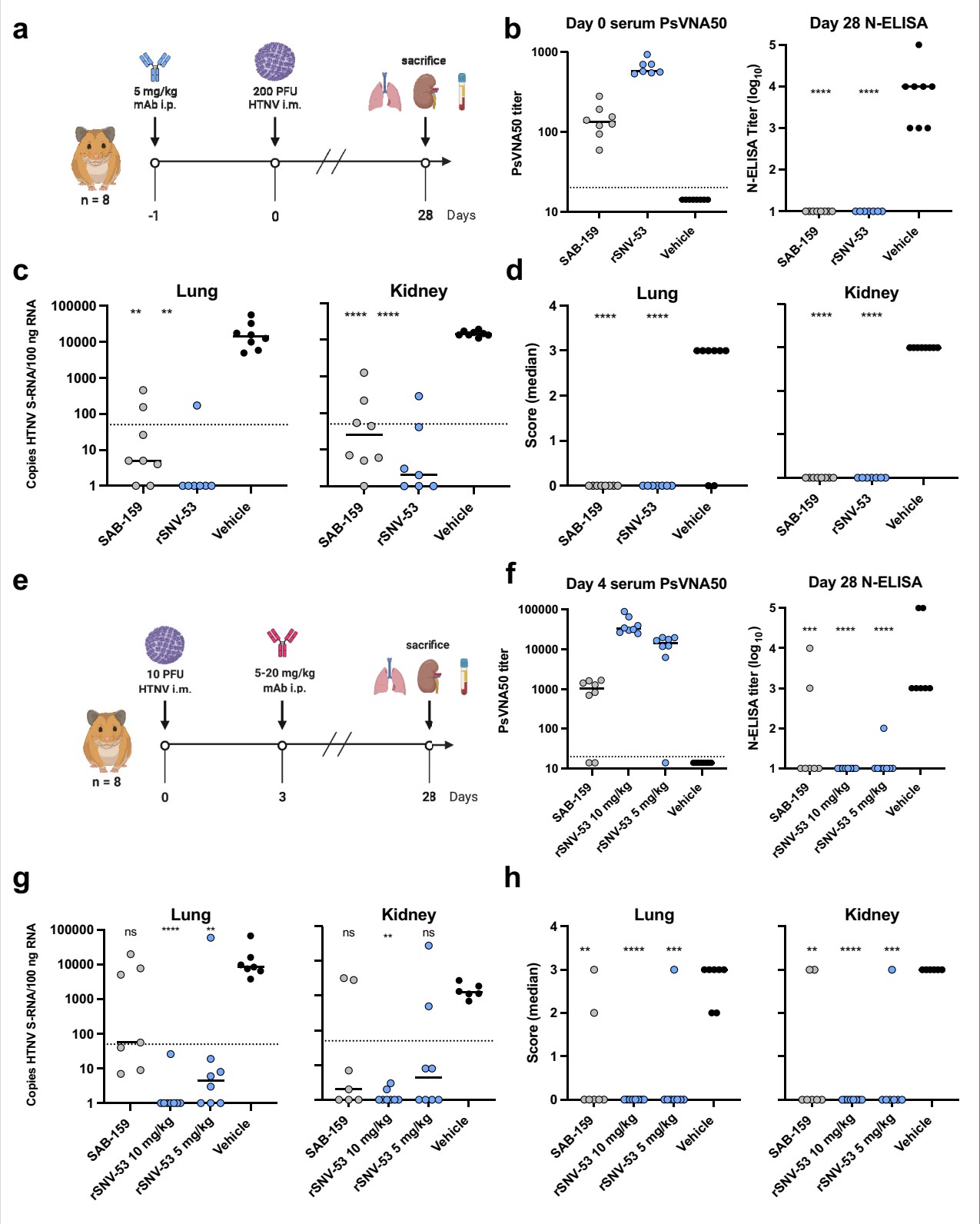

**Figure 7.** SNV-53 protects hamsters when given before or after HTNV inoculation. (**a**) 8-week-old Syrian hamsters (n=8 per treatment group) were administered 5 mg/kg of the indicated Ab treatment and then inoculated 1 day later with 200 PFU of HTNV i.m. All animals were sacrificed at 28 dpi and organs were harvested. (**b**) Ab detection in the serum at day 0 by pseudovirion neutralization assay (PsVNA) and at day 28 by nucleoprotein ELISA. Dotted line indicated limit of detection. (**c**) qRT-PCR detection of HTNV genome in the lungs and kidneys. Kruskal-Wallis test with multiple comparisons

*Figure 7 continued on next page*

*Figure 7 continued*

of each group to vehicle, * p<0.01, ** p<0.001, *** p<0.001, **** p<0.0001. ns, not significant. Dashed line indicates the limit of detection, which was 50 copies of HTNV S-RNA per 100 ng/100 ng RNA.(**d**) In-situ hybridization detection of HTNV genome in the lungs and kidneys. One-way ANOVA with multiple comparisons of each group to vehicle, * p<0.01, ** p<0.001, *** p<0.001, **** p<0.0001. ns, not significant. (**e**) Eight-week-old Syrian hamsters (n=6 per treatment group) were inoculated with 10 PFU of HTNV i.m., and 5–10 mg/kg of indicated Ab was administered via i.p. route at 3 dpi. All animals were sacrificed at 28 dpi and organs were harvested. (**f**) Serum was analyzed as indicated in **b**. (**g**) Lung and kidneys were assayed as described in **c**.(**h**) Lung and kidneys were assayed as described in **d**.

The online version of this article includes the following figure supplement(s) for figure 7:

**Figure supplement 1.** Pre- and post-exposure HTNV serum PsVNA50 data and pathology staining.

vaccines, respectively, and previously were demonstrated to protect animals in pre-and post-exposure models of HTNV and PUUV infection (***Perley et al., 2020***). Day 0 bioavailability of neutralizing antibody measured through a pseudovirion neutralization assay (PsVNA) indicated that injected antibody was present in sera as expected for the positive control antibody and SNV-53 (***Figure 7b***). PsVNA levels measured on day 28 after virus inoculation revealed that all vehicle-control-treated animals had seroconverted in response to productive infection, while all animals injected with SAB-159 or SNV-53 lacked an increase in PsVNA activity on day 28 (***Figure 7—figure supplement 1***). Day 28 ELISA for antibodies binding to nucleocapsid (N) protein confirmed that animals injected with SAB-159 or SNV-53 did not seroconvert (***Figure 7b***).

As expected, pathologic changes in the tissue of the infected hamsters were not detected by hematoxylin and eosin staining; however, a remarkably high amount of virus genome was detected in multiple organs by qRT-PCR and in situ hybridization (ISH). Animals injected with SAB-159 or SNV-53 had significantly lower levels of viral RNA in the lungs and kidneys compared to vehicle control-treated animals (***Figure 7c***). Lungs and kidneys for all animals injected with SAB-159 or SNV-53 were negative for genome by ISH, whereas all animals injected with vehicle were positive by ISH for genome in both lungs and kidneys (***Figure 7d***). Many of the genome-positive organs had large amounts of staining (score 3) despite the presence of neutralizing and anti-N antibodies in the serum that were mostly IgG (***Figure 7—figure supplement 1***).

## SNV-53 MAb cross-protects against an Old World hantavirus when administered after virus inoculation

We next tested SNV-53 in a post-exposure model of HTNV infection. Hamsters were inoculated with 10 PFU HTNV by the i.m. route and then treated with SNV-53 at 10 or 5 mg/kg, the positive control SAB-159, or a negative control mAb DENV 2D22 at day 3 post-exposure, and then sacrificed 28 days later and lungs, kidneys, and serum were harvested (***Figure 7e***). Day 4 post-infection (1 day after antibody injection) bioavailability studies confirmed presence of SAB-159 or SNV-53 in sera (***Figure 7f***). All animals with PsVNA$_{50}$ serum titers lower than ~200 developed higher serum titers on day 28, indicating the animals had responded to the infection (***Figure 7—figure supplement 1***). Day 28 N-ELISA studies confirmed that post-exposure injection of animals with SAB-159 or SNV-53 significantly reduced seroconversion (***Figure 7f***). The most significant impact was when SNV-53 was injected, and there appeared to be a dose-response in which SNV-53 given at a 10 mg/kg dose was most protective. Only 1 of the 16 animals injected with SNV-53 antibody had detectable serum anti-N antibodies on day 28.

qRT-PCR testing detected high levels (>1000 copies/500 ng RNA) of virus RNA in the lungs of all the control-treated hamsters (***Figure 7g***). There was a statistically significant reduction in the RNA levels in the lungs of all SNV-53-treated groups. There was a similar readout in the kidneys, except that the SNV-53 protection at a 5 mg/kg dose was not statistically significant.

Five of the seven hamsters in the positive control-treated group had no detectable virus RNA as measured by ISH in lungs or kidneys on day 28, which was a significant reduction relative to the vehicle-control-treated group (***Figure 7h***). The level of protection for both doses was significant in the lung but not the kidney. All animals injected with the 10 mg/kg dose and all but one animal injected with the 5 mg/mL of SNV-53 had no virus RNA detected in the lungs or kidneys. The one positive animal in the SNV-53 with a high level of virus in both lung and kidney was the same animal that did not receive the correct antibody dose as measured PsVNA on day-4 sera (***Figure 7f***).

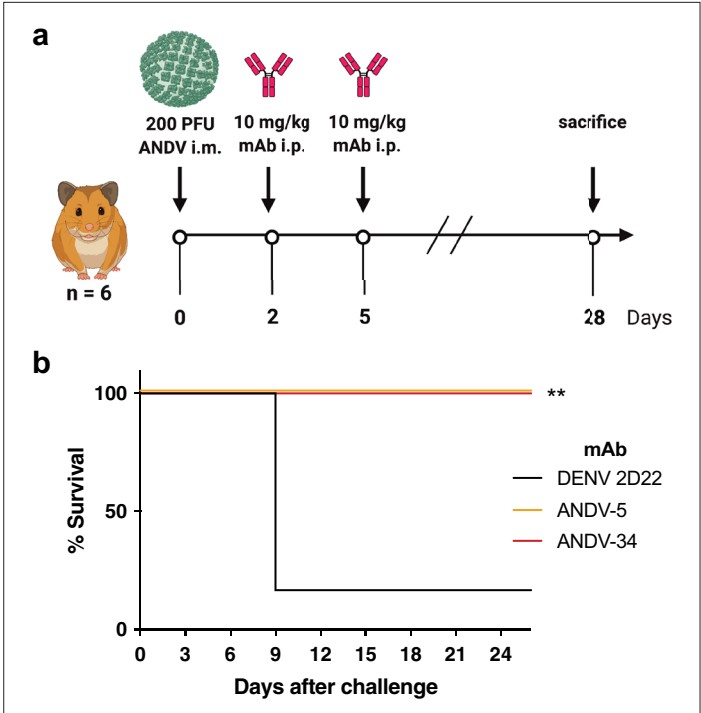

**Figure 8.** ANDV-34 and ANDV-5 protect Syrian golden hamsters for lethal ANDV challenge. (**a**) Eight-week-old Syrian hamsters (n=6 per treatment group) were inoculated with 200 PFU of ANDV i.m., and 10 mg/kg of indicated mAb was administered via i.p. route at 2 and 5 dpi. Animals were treated with a dengue-specific mAb (DENV 2D22) to serve as an isotype control. (**b**) Kaplan-Meier survival plot. Statistical analysis of survival curves was done using a log-rank (Mantel-Cox) test comparing each group to the control (DENV 2D22), **, p=0.0014.

## ANDV-specific antibodies protect from ANDV challenge in hamsters

We previously demonstrated that ANDV-5 was partially protective against lethal disease in hamsters after ANDV infection (*Engdahl et al., 2021*). Previous studies have shown that inoculation of Syrian golden hamsters with ANDV elicits a pathological response like that seen during HCPS in humans (*Hooper et al., 2001b*). To achieve complete protection and test ANDV-34 for therapeutic efficacy in this model, Syrian golden hamsters were inoculated with 200 PFU of ANDV virus (strain Chile-9717869) and administered 10 mg/kg of ANDV-5, ANDV-34, or DENV-2D22 (isotype control) on days 2 and 5 post-inoculation (*Figure 8a*). Both ANDV-5 and ANDV-34 demonstrated complete protection from disease and mortality, whereas all but one of the isotype control-treated animals succumbed on day 9 (*Figure 8b*). Thus, antibodies elicited to both antigenic sites on Gn$^H$ can provide protection from lethal HCPS-like disease in vivo.

## Discussion

Antibodies isolated from survivors of hantavirus infection have been shown to reduce viral replication and protect against disease in animal models (*Mittler et al., 2022*; *Engdahl et al., 2021*; *Garrido et al., 2018*), suggesting that mAbs might be effective for preventing or treating hantavirus-related disease in humans. The selection of optimal lead candidate antibodies for monotherapy or rational composition of an antibody combination requires detailed information on the antigenic sites of vulnerability to inhibition. Here, we defined four major antigenic sites on the hantavirus Gn/Gc complex recognized by neutralizing and protective antibodies. The antibodies are promising for development as biologics, and the landscape analysis of protective sites that emerges from the mapping studies provides a blueprint for virus species-specific or broadly protective vaccine design.

Two classes of broad antibodies we describe exhibit breadth and likely neutralize through viral fusion inhibition. These antibodies are strikingly similar to group I and group II bnAbs isolated from PUUV convalescent donors (*Mittler et al., 2022*), demonstrating the immunodominance of these

antigenic sites recognized by the human humoral repertoire. Apart from flaviviruses, canonical class II fusions proteins are hidden in the three-dimensional structure of the glycoprotein spike by an accompanying protein to prevent premature fusion during viral entry (*Guardado-Calvo and Rey, 2021b*). Once viral attachment occurs, the virion is taken up in the endosome, where low-pH triggers the dissociation of the complex and exposes the hydrophobic fusion loop initiating viral-host cell fusion and entry into the cytoplasm. We propose that one class of hantavirus bnAbs we have studied here (i.e. SNV-53 and ANDV-44) stabilize the association of Gn (the accompanying protein) to Gc (the fusion protein) by binding to and cross-linking the two subunits – preventing the heterodimer from undergoing conformational changes necessary for fusion.

BnAbs that span the heterodimeric interface also have been described for other enveloped viruses, including hepatitis C virus (HCV; *Colbert et al., 2019*), Rift Valley Fever virus (RVFV; *Chapman et al., 2021*), and chikungunya virus (*Zhou et al., 2020*). One antibody, RVFV-140, isolated from a Rift Valley Fever virus survivor, potently neutralizes RVFV through a fusion inhibition mechanism and targets an epitope only present on the Gn/Gc hetero-hexamer (*Chapman et al., 2021*). Interface targeting antibodies can aid in the structural determination of heterodimers; the structure of the hepatitis C virus E1E2 heterodimer was determined by co-expressing the complex with a bnAb, AR4A, which targets an epitope proximal to the E1E2 interface (*de la Peña et al., 2021*). Altogether, the data indicate that there exists an essential class of bnAbs that target quaternary epitopes at the interface regions on heterodimers for many viruses. These antibodies appear to be induced by viruses in diverse families that use heterodimeric surface glycoproteins, and the shared mechanism appears to be fusion inhibition by preventing the complex conformational changes required for viral fusion. The structures of the antigenic sites recognized by these rare antibodies could provide a blueprint for structure-based reverse vaccinology efforts to design stabilized viral immunogens to elicit a broad, protective antibody response (*Byrne and McLellan, 2022*).

SNV-53 did cross-protect both in prophylactic and therapeutic small animal models of HTNV and ANDV infection, suggesting that SNV-53 is a promising candidate for treating both HCPS and HFRS. Based on studies using germline revertant forms of these antibodies, SNV-53 and ANDV-44 have developed their breadth and neutralization potency across multiple species through the accumulation of somatic mutations. It may be possible to continue to affinity mature SNV-53-like antibodies to increase the potency and breadth across species. Combined with the recent description of Gn/Gc interface mAbs from PUUV convalescent donors (*Mittler et al., 2022*), our findings indicate that these types of antibodies can be readily elicited during natural infection and are likely critical to pan-hantavirus therapeutic and vaccine development.

The second class of bnAbs, exemplified by SNV-24, targets the domain I region on Gc. This region is highly conserved, which likely explains why SNV-24 demonstrates broad reactivity. The putative epitope for SNV-24 is near the epitope identified for bank vole-derived antibodies 1C9 and 4G2, and a putative mechanism for this antibody class is the inhibition of viral fusion through blocking the formation of a post-fusion trimer (*Rissanen et al., 2020*; *Lundkvist and Niklasson, 1992*). However, this site contacts neighboring Gc protomers, therefore, it may not be fully accessible on the lattice virion surface. These accessibility considerations may contribute to the observed incomplete neutralization of authentic hantaviruses by SNV-24 (*Mittler et al., 2022*). SNV-24 does potently neutralize hantavirus glycoprotein pseudotyped VSVs, but this finding might be due to an alternative arrangement of the glycoprotein spikes on the VSV virion that does not correctly mimic the glycoprotein array on authentic viruses.

Although understanding bnAbs is important, ANDV-specific nAbs will still be critical to deploying medical countermeasures. ANDV represents the only hantavirus shown to transmit person-to-person and causes notable outbreaks yearly in South America (*Martínez et al., 2020*). Potently neutralizing ANDV antibodies – represented by ANDV-5 and ANDV-34 – target two distinct faces of the globular, membrane distal Gn head domain. The epitope for ANDV-5 maps to the α–3 and α–4 helices, while ANDV-34 targets domain B. Both sites of vulnerability do not overlap with the epitopes determined for previously described Gn-targeting mAbs, HTN-Gn1 and nnHTN-Gn2 (*Rissanen et al., 2017*; *Rissanen et al., 2021*). A canonical neutralization mechanism used by potent antibodies is to block the attachment of viruses to host cells. Here, we demonstrate that selected ANDV-neutralizing human mAbs block virus binding to the receptor for hantavirus entry protocadherin-1 (PCDH-1), specifically to the first extracellular (EC-1) domain amino-acid (residues 61–172) or EC-2 domain (residues

173–284) (*Jangra et al., 2018*). The finding that ANDV-specific, potently neutralizing mAbs function through PCDH-1 blockade is consistent with the finding that PCDH-1 is used by NWHs but not Old World hantaviruses (*Jangra et al., 2018*). Although the RBS on hantavirus glycoproteins is currently unknown, we propose that the site likely overlaps with the epitope of ANDV-5 since the Fab form of ANDV-5 also competes with sEC1-EC2 indicating that steric clashes are the not the primary driver of receptor blocking (*Mittler et al., 2019*). However, it also possible that the RBS spans two Gn:Gn protomers, as we demonstrated that ANDV-5 Fab may contact neighboring, intraspike Gn subunits. Both antibodies protect against HCPS-like disease in Syrian hamsters, indicating that both epitopes on the Gn can elicit a protective neutralizing antibody response.

We also show that JL16 can be categorized in a similar antibody class as ANDV-5, while MIB22 can be classified into a similar class as ANDV-34. Notably, three ANDV-specific nAbs (ANDV-5, ANDV-34, and MIB22) from two different human donors are encoded by *IGHV1-69* and use the F alleles *01* and *06*. IGHV1-69* encoded antibodies are prevalent in response to many different pathogens, including influenza (*Pappas et al., 2014*), HIV (*Zhou et al., 2015*), HCV (*Bailey et al., 2017*), EBOV (*Murin et al., 2021*), and even *Staph aureus* (*Bennett et al., 2019*). $V_H$1-69 gene usage is overrepresented in antiviral antibodies due to the genetic characteristics of the germline-encoded CDRH2 loop (*Chen et al., 2019*). Rational vaccine design should promote the formation of $V_H$1-69 antibodies since they have a low somatic hypermutation barrier to high-affinity binding and can be readily elicited in most individuals. Hydrophobic interactions facilitated by the templated amino acids in the CDHR2 loop may contribute to recognizing ANDV Gn by naïve B cells during acute infection.

RNA viruses are particularly prone to mutations and rapid viral evolution, therefore, prospective identification of escape mutations and defining the likelihood of escape is an essential step in developing antibody combinations. Strikingly, five of the nAbs tested here were particularly resistant to escape, and most escape mutants eventually selected through serial passaging had multiple mutations. This finding could indicate that a single mutation is insufficient to confer antibody escape while maintaining the functionality required for viral fitness. Although it was challenging to generate neutralization-resistant variants, we did show that some single mutations could completely ablate the binding of multiple antibodies, even if the mutation was located distant from the putative binding site in the three-dimensional structure. It has been previously demonstrated in other viruses, such as SARS-CoV-2, that single mutations can have functional consequences on the 3D structure of antigenic proteins, thereby altering antibody activity. For example, S371L confers resistance to numerous SARS-CoV-2 antibodies from multiple classes (*Liu et al., 2022*), while E406W causes allosteric remodeling of the receptor-binding domain that significantly impacts the binding of the REGEN-CoV therapeutic monoclonal antibody cocktail (*Addetia et al., 2022*; *Starr et al., 2021*). Using antibody combinations that resist escape from single-point mutations is an important objective. MIB22 and JL16 have demonstrated efficacy as a monotherapy and a combination therapy in vivo (*Garrido et al., 2018*; *Williamson et al., 2021*). It is not clear the benefit of combining these two mAbs, since they partially compete for binding, have a similar mechanism of action, and are vulnerable to escape from similar amino acid changes (S309Y/D336Y).

There is still much to learn about the humoral immune response to the hantavirus Gn and Gc proteins, including where major functional epitopes are. Here, we begin to unravel the epitopes targeted by neutralizing antibodies elicited during infection with New World hantaviruses. This endeavor is critical for vaccine design against numerous pathogens, including RSV (*Graham et al., 2015*), HIV (*Kwong and Mascola, 2018*), and, more recently, SARS-CoV-2 (*Barnes et al., 2020*). To prepare for the possible emergence of novel hantaviruses, it is imperative to understand the functional antigenic sites on the glycoprotein spike that need to be presented to the immune response to generate an effective vaccine.

# Materials and methods
## Cell lines
Expi293F cells (Thermo Fisher Scientific, Cat# A14527; RRID:CVCL D615, female) were cultured in suspension at 37 °C in 8% $CO_2$ shaking at 125 RPM in Freestyle F17 Expression Medium (GIBCO Cat# A13835-01) supplemented with 10% Pluronic F-68 and 200 mM of L-glutamine. ExpiCHO cells (Thermo Fisher Scientific Cat# A29127; RRID:CVCL 5J31, female) were cultured in suspension at 37 °C

in 8.0% $CO_2$ shaking at 125 RPM in ExpiCHO Expression Medium (Thermo Fisher Scientific). Vero cell lines (ATCC, CCL-81, African green monkey, female) were cultured at 37 °C in 5% $CO_2$ in DMEM (Thermo Fisher) supplemented with 10% fetal bovine serum. *Drosophila* S2 cells (Thermo Fisher Scientific Cat# 51–4003) were cultured in suspension at 37 °C shaking at 125 RPM in Schneider's *Drosophila* Medium (Thermo Fisher Scientific, Cat# 21720001) supplemented with 10% fetal bovine serum. All cell lines were tested for mycoplasma monthly, and all samples were negative.

## Viruses

Replication-competent, recombinant VSV strains bearing ANDV or SNV glycoproteins were kindly provided by K. Chandran and propagated on Vero cells as previously described (*Jangra et al., 2018*). HTNV 76–118 (*Lee et al., 1978*) was plaque purified twice and passaged in Vero E6 cells, as previously described (*Hooper et al., 2001a*). Andes virus strain Chile-9717869 (Chile R123) were obtained from the World Reference Center for Emerging Viruses and Arboviruses housed at UTMB.

## Plasmids

Plasmids containing a cDNA encoding the full-length M segment from SNV [pWRG/SN-M(opt)] (*Hooper et al., 2013*) and ANDV [pWRG/AND-M(opt2)] (*Hooper et al., 2014*), were used to produce cell-surface displayed hantavirus antigens. Plasmids used to produce stable S2 cell lines expressing recombinant hantavirus antigens (pT350-MAPV Gn^H/Gc, pT350-ANDV Gn^H/Gc, pT350-ANDV Gn^H/Gc (H953F), pT350-ANDV Gn^B, and pT350-ANDV Gc) were provided by Dr. Felix Rey and Dr. Pablo Guardado-Calvo, Institut Pasteur. The cDNAs encoding antibody genes were synthesized by Twist Biosciences.

## Recombinant human IgG1 and Fab expression and purification

ExpiCHO cells were transiently transfected using the ExpiCHO Expression System (GIBCO) with plasmids encoding human IgG1 or Fab cDNAs. Supernatant was harvested from ExpiCHO cultures and filtered with 0.45 mm pore size filter flasks. HiTrap MabSelectSure columns (Cytiva) or Capture-SelectTM CH1-XL columns (Thermo Fisher Scientific) were used to affinity purify IgG1 or Fab from ExpiCHO supernatant using an ÄKTA pure protein purification system (Cytiva).

## Expression and purification of hantavirus antigens

Soluble, recombinant hantavirus antigens were expressed and purified as previously described (*Serris et al., 2020*). Genes coding for the ANDV Gn head domain (GenBank: NP_604472.1, residues 21–374) and the Gc ectodomain (GenBank: NP_604472.1, residues 652–1107) were combined into a single chain using a flexible linker (GGSGLVPRGSGGGSGGGSWSHPQFEKGGGTGGGTLVPRGSGTGG), and single domain ANDV constructs for Gn base domain (Gn^B, residues 375–484), Gn head (Gn^H, residues 21–374), and Gc (residues 652–1107) were also generated. A similar linked construct was made for Maporal virus (MAPV, strain HV-97021050, NCBI code YP_009362281.1). The constructs were codon-optimized for *Drosophila* cell expression and cloned into a plasmid (pT350) containing an MT promoter, BiP signal sequence, and C-terminal double strep tag (GenScript, kindly provided by P. Guardado-Calvo and F. Rey). *Drosophila* S2 cells were transfected with the pT350-ANDV Gn^H/Gc and pCoBlast (Thermo Fisher Scientific) plasmids at a 19:1 ratio, respectively, and stably-transfected cells were selected using 25 µg/mL of blasticidin. Stable cell lines were maintained in Schneider's *Drosophila* Medium supplemented with 25 µg/mL blasticidin and grown in shaker flasks to a density of 1x10^7 cells/mL and induced with 4 µM $CdCl_2$. S2 cell supernatant was collected after 5 days and supplemented with 10 µg/mL of avidin and purified through a StrepTrap HP column (Cytiva). The sample was purified further by size-exclusion chromatography on HiLoad 16/600 Superdex column (Cytiva).

## Negative-stain electron microscopy

Electron microscopy imaging was performed, as previously described (*Doyle et al., 2021*; *Zost et al., 2020*), with ANDV Gn^H protein in complex with ANDV-5 and ANDV-34 and MAPV Gn^H/Gc in complex with SNV-53 and SNV-24. Recombinant forms of ANDV Gn^H and MAPV Gn^H/Gc were expressed and purified as described above. Fabs forms of ANDV-5, ANDV-34, SNV-53, and SNV-24 were expressed and purified as described above. Complexes were generated by incubating the recombinant proteins

with the two corresponding Fabs in a 1:1.2:1.2 (antigen:Fab:Fab) molar ratio. 3 μL of the sample at ~10 μg/mL was applied to a glow-discharged grid with continuous carbon film on 400 square mesh copper electron microscopy grids (Electron Microscopy Sciences). Grids were stained with 2% uranylformate (*Ohi et al., 2004*). Images were recorded on a Gatan US4000 4kX4k CCD camera using an FEI TF20 (TFS) transmission electron microscope operated at 200 keV and control with SerialEM (*Mastronarde, 2005*). All images were taken at 50,000 magnification with a pixel size of 2.18 Å/pixel in low-dose mode at a defocus of 1.5–1.8 mm. The total dose for the micrographs was 33 e/Å2. Image processing was performed using the cryoSPARC software package (*Punjani et al., 2017*). Images were imported, CTF-estimated and particles were picked automatically. The particles were extracted with a box size of 180 pix and binned to 96 pix (4.0875 Å/pixel) and multiple rounds of 2D class averages were performed to achieve clean datasets. The final dataset was used to generate Initial 3D volume and the volume was refine for final map at the resolution of ~18 Å. Model docking to the EM map was done in Chimera (*Pettersen et al., 2004*). For the ANDV $Gn^H$/Gc protomer or tetramer PDB: 6Y5F or 6ZJM were used and for the Fab PDB:12E8. Chimera or ChimeraX software was used to make all Figures, and the EM map has been deposited into EMDB (EMD-26735 and EMD-26736).

## Cryo-EM sample preparation and data collection

Two Fabs ANDV- 5, ANDV-34 and ANDV $Gn^H$ were expressed recombinantly and combined in a molar ration of 1:1.2:1.2 (Ag:Fab:Fab). The mixture was incubated for over-night at 4 °C and purified by gel filtration. 2.2 μl of the purified mixture at concentration of 0.2 mg/mL was applied to glow discharged (40 s at 25mA) UltraAu grid (300 mesh 1.2/1.3, Quantifoil). The grids were blotted for 3 s before plunging into liquid ethane using Vitrobot MK4 (TFS) at 20 °C and 100% RH. Grids were screened and imaged on a Glacios (TFS) microscope operated at 200 keV equipped with a Falcon 4 (TFS) DED detector using counting mode and EEF. Movies were collected at nominal magnification of ×130,000, pixel size of 0.73 Å/pixel and defocus range of 0.8–1.8 μm. Grids were exposed at ~1.18 e/aÅ2/frame resulting in total dose of ~50 e/Å2 (*Figure 4—figure supplement 1* and *Supplementary file 1*).

## Cryo-EM data processing

Data processing was performed with Relion 4.0 beta2 (*Kimanius et al., 2021*). EER Movies were preprocessed with Relion Motioncor2 (*Zheng et al., 2017*) and CTFFind4 (*Rohou and Grigorieff, 2015*). Micrographs with low resolution, high astigmatism and defocus were removed from the data set. The data set was autopicked first by Relion LoG (*Fernandez-Leiro and Scheres, 2017*) and was subject to 2D classification. Good classes were selected and used for another round of autopicking with Topaz training and Topaz picking (*Bepler et al., 2020*). The particles were extracted in a box size of 180 pixel and binned to 128 pixels (pixel size of 2.053 Å/pixel). The particles were subjected to multiple rounds of 2D class averages, 3D initial map and 3D classification to obtain a clean homogeneous particle set. This set was re-extracted at a pixel size of 1.46 Å/pixel and was subjected to 3D autorefinement. The data were further processed with CTFrefine *Zivanov et al., 2018*, polished and subjected to final 3D autorefinement and postprocessing. Detailed statistics are provided in *Figure 4—figure supplement 1* and *Supplementary file 1*.

## Model building and refinement

For model building PDB: 6Y5F9 was used for $Gn^H$. Three AlphaFlod2 predictions for the Fv of ANDV-5, Fv of ANDV-34 or Fc segment of the Fab was used for the Fabs. All the models were first docked to the map with Chimera (*Pettersen et al., 2004*) or ChimeraX (*Pettersen et al., 2021*). The $Gn^H$ and the Fc were rigid-body real-space refined with Phenix (*Adams et al., 2010*). To improve the coordinates of ANDV-5 and ANDV-34 Fv, the models were subjected to iterative refinement of manual building in Coot (*Emsley and Cowtan, 2004*) and Phenix. The models were validated with Molprobity (*Chen et al., 2010*; *Supplementary file 1*).

## Bio-layer interferometry competition binding analysis

An Octet Red96 instrument (Sartorius) was used for competition binding analysis of antibodies. Octet Streptavidin (SA) Biosensors (Sartorius, Cat # 18–5019) were incubated in 200 μL of 1×kinetics buffer (diluted in D-PBS) for 10 min. Baseline measurements were taken in 1×kinetics buffer for 60 s and then recombinant ANDV $Gn^H$/Gc (10 μg/mL) was loaded onto the tips for 180 s. The tips were washed for

30 s in 1×kinetics buffer and then the first antibody (50 µg/mL) was associated for 300 s. The tips were washed again for 30 s in 1×kinetics buffer and then the second antibody (50 µg/mL) was associated for 300 s. The maximal binding of each antibody was calculated by normalizing to the buffer only control. The Octet epitope software was used to determine the percent competition for each antibody and antibodies were clustered using the Pearson correlation.

### ELISA binding assays

384-well plates were coated with 1 µg/mL of purified recombinant protein overnight at 4 °C in carbonate buffer (0.03 M $NaHCO_3$, 0.01 M $Na_2CO_3$, pH 9.7). Plates were incubated with blocking buffer (2% non-fat dry milk, 2% goat serum in PBS-T) for 1 hr at room temperature. Serially-diluted primary antibodies were added to appropriate wells and incubated for 1 hr at room temperature, and horseradish peroxidase-conjugated secondary antibodies (Southern Biotech) were used to detect binding. TMB substrate (Thermo Fisher Scientific) was added, and plates were developed for 5 min before quenching with 1 N hydrochloric acid and reading absorbance at 450 nm on a BioTek micro-plate reader.

### Escape mutant generation through serial passaging

VSV/ANDV or VSV/SNV escape mutants were generated by serially passaging the viruses in Vero cell monolayer cultures with increasing concentrations of SNV-53, SNV-24, or ANDV-44. VSV/ANDV or VSV/SNV (MOI of ~1) were incubated for 1 hr at 37 °C with each antibody at their respective $IC_{50}$ value, and the mixture was added to Vero cells in a 12-well plate. Cells were monitored for CPE and GFP expression starting at 72 hr after inoculation to confirm virus propagation. Once ~90% of the cells were GFP+. A+200 µL volume of supernatant containing the selected virus was passaged to a fresh Vero monolayer in the presence of increasing concentrations of the selection antibody. After 6–7 passages, escape mutant viruses were propagated in a six-well culture plate in the presence of 10 µg/mL of the selection antibody. Viruses were also propagated in the presence of 10 µg/mL of the non-selection antibodies to assess susceptibility. Viral RNA was then isolated using a QiAmp Viral RNA extraction kit (QIAGEN) from the supernatant containing the selected viral mutant population. The ANDV or SNV M gene cDNA was amplified with a SuperScript IV One-Step RT-PCR kit (Thermo Fisher Scientific) using primers flanking the M gene. The resulting amplicon (~3500 bp)was purified using SPRI magnetic beads (Beckman Coulter) at a 1:1 volume ratio and sequenced by the Sanger sequence technique using primers giving forward and reverse reads of the entire M segment. The full-length M gene segments for the original VSV/SNV and VSV/ANDV stock used was also confirmed through Sanger sequencing. Each read was aligned to the VSV/ANDV or VSV/SNV genome to identify mutations.

### Site-directed mutagenesis of M segment gene

Plasmids containing a cDNA encoding the full-length M segment from SNV pWRG/SN-M(opt) (*Hooper et al., 2013*) and pWRG/AND-M(opt2) (*Hooper et al., 2014*) were used mutagenized using the Q5 Site-Directed Mutagenesis Kit (NEB, cat no. E0554S). Primers were designed using the NEBaseChanger tool (https://nebasechanger.neb.com/), and PCR reactions were performed according to the manufacturer's manual. Plasmids were transformed into NEB 5-alpha Competent *E. coli* cells (New England Biolabs; cat no. C2987I). All mutants were sequenced using Sanger DNA sequencing to confirm the indicated nucleotide changes.

### Expression of mutant M segments

For expression of mutant constructs, we performed micro-scale transfection (700 µL) of Expi293F cell cultures using the Gibco ExpiFectamine 293 Transfection Kit (Thermo Fischer Scientific, cat no. A14525) and a protocol for deep 96-well blocks (Nest, cat no. 503162). Briefly, sequence confirmed plasmid DNA was diluted in OptiMEM I serum-free medium and incubated for 20 min with ExpiFectamine 293 Reagent (Gibco). The DNA– ExpiFectamine 293 Reagent complexes were used to transfect Expi293F cell cultures in 96-deep-well blocks. Cells were incubated, shaking at 1000–1500 rpm inside a humidified 37 °C tissue culture incubator in 8% $CO_2$ and harvested 48 hr post-transfection.

### Flow cytometric binding analysis to mutant M segments

A flow cytometric assay was used to screen for and quantify binding of antibodies to the mutant constructs. Transfected Expi293F were plated into 96-well V-bottom plates at 50,000 cells/well in

FACS buffer (2% ultra-low IgG FBS, 1 mM EDTA, D-PBS). Hantavirus-specific mAbs or an irrelevant mAb negative control, rDENV-2D22, were diluted to 1 µg/mL in FACS buffer and incubated 1 hr at 4 °C, washed twice with FACS buffer, and stained with a 1:1000 solution of goat anti-human IgG antibodies conjugated to phycoerythrin (PE) (Southern Biotech, cat no. 2040–05) at 4 °C for 30 min. Cells then were washed twice with FACS buffer and stained for 5 min at 4 °C with 0.5 mg/mL of 40,6-diamidino-2-phenylindole (DAPI) (Thermo Fisher). PE and DAPI staining were measured with an iQue Screener Plus flow cytometer (Intellicyt) and quantified using the manufacturer's ForeCyt software. A value for percent PE-positive cells was determined by gating based on the relative fluorescence intensity of mock transfected cells Tab.

## Real-time cell analysis (RTCA) neutralization assay

A high-throughput RTCA assay that quantifies virus-induced cytopathic effect (CPE) was used to test antibody-mediate virus neutralization under BSL-2 conditions, using a general approach previously described for other viruses (*Gilchuk et al., 2020*). VSV/ANDV and VSV/SNV were titrated by RTCA on Vero cell culture monolayers to determine the concentration that elicited complete cytopathic effect (CPE) at 36 hr post-inoculation. Fifty µL of Dulbecco's Modified Eagle Medium (DMEM) supplemented with 2% FBS was added to each well of a 96-well E-plate (Agilent) to establish the background reading, then 50 µL of Vero cells in suspension (18,000 cells/well) were seeded into each well to adhere. Plates were incubated at RT for 20 min, and then placed on the xCELLigence RTCA analyzer (formerly ACEA Biosciences, now Agilent). Cellular impedance was measured every 15 min. VSV/ANDV (~1000 infectious units [IU] per well) or VSV/SNV (~1200 IU per well) were mixed 1:1 with mAb in duplicate in a total volume of 100 µL using DMEM supplemented with 2% FBS and incubated for 1 hr at 37 °C in 5% $CO_2$. At 16–18 hr after seeding the cells, the virus-mAb mixtures were added to the cells. Wells containing virus only (no mAb added) or cells only (no virus or mAb added) were included as controls. Plates were measured every 15 min for 48 hr post-inoculation to assess for inhibition of CPE as a marker of virus neutralization. Cellular index (CI) values at the endpoint (36 hr after virus inoculation) were determined using the RTCA software version 2.1.0 (Agilent). Percent neutralization was calculated as the CI in the presence of mAb divided by the cells only (no-CPE control) wells. Background was subtracted using the virus only (maximum CPE) control wells. Half maximal inhibitory concentration ($IC_{50}$) values were calculated by nonlinear regression analysis using Prism software version 9 (GraphPad).

## RTCA single-passage escape mutant screening

The RTCA method described above was modified to generate escape mutant viruses after a single passage under saturating neutralizing antibody concentrations (*Greaney et al., 2021*; *Suryadevara et al., 2022*; *Gilchuk et al., 2021*). 0.5, 1, 5, 10, or 20 µg/mL (depending on the potency of the antibody to the given virus) of the selection antibody was mixed 1:1 in 2% FBS-supplemented DMEM with VSV/ANDV (~10,000 IU per well) or VSV/SNV (~12,000 IU per well) and incubated for 1 hr at 37 °C and then added to cells. Virus only (no mAb) and medium only (no virus) wells were included as controls. Escape mutant viruses were identified by a drop in cellular impedance over 96 hr. Supernatants from wells containing neutralization resistant viruses were expanded to 12-well plates in saturating concentrations of the selection antibody to confirm escape-resistant phenotype and the presence of other neutralizing mAbs as a control. Supernatants were filtered and stored at –80 °C.

## Hydrogen-deuterium exchange mass spectrometry (HDX-MS)

ANDV GnH-Gc protein and Fabs were prepared at 15 pmol/µL. Labeling was done in Dulbecco's phosphate-buffered saline (DPBS), pH 7.4, in $H_2O$, pH 7.4 (no labelling) or $D_2O$, pH 7.0 (labelling). Samples were incubated for 0, 10, 100, 1000, or 5000 s at 20 °C. The labelling reaction was quenched by addition of 50 µL TCEP quench buffer, pH 2.4, at 0 °C. Automated HDX incubations, quenches, and injections were performed using an HDX-specialized nano-ACQUITY UPLC ultraperformance liquid chromatography (UPLC) system coupled to a Xevo G2-XS mass spectrometer. Online digestion was performed at 20 °C and 11.600 psi at a flow of 150 µL/min of 0.1% formic acid in $H_2O$ (3.700 psi at 40 µL/min), using an immobilized-pepsin column. Peptides were identified in un-deuterated samples using Waters ProteinLynx Global Server 3.0.3 software with non-specific proteases, minimum fragment ion matches per peptide of three and oxidation of methionine as a variable modification. Deuterium uptake was calculated and compared to the non-deuterated sample using DynamX 3.0 software.

Criteria were set to a minimum intensity of 5000, minimum products 4 and a sequence length 5–25 residues.

## Fusion from without (FFWO) assay

A FFWO assay that measures fusion-dependent antibody neutralization was used as previously described for other bunyaviruses (*Chapman et al., 2021*). Sterile, cell-culture-treated 96-well plates were coated overnight at 4 °C with 50 µg/mL of poly-d-lysine (Thermo Fisher Scientific). Coated plates were washed with D-PBS and dried, and Vero cells were plated at 18,000 cells/well and incubated at 37 °C in 5.0% $CO_2$ for 16 hr. The supernatant was aspirated, and cells were washed with binding buffer (RPMI 1640, 0.2% BSA, 10 mM HEPES pH 7.4, and 20 mM $NH_4Cl$) and incubated at 4 °C for 15 min. VSV/ANDV was diluted to an MOI of ~1 in binding buffer and added to cells for 45 min at 4 °C. Cells were washed to remove unbound viral particles. Hantavirus-specific or control mAbs were diluted in DMEM supplemented with 2% FBS and added to cells for 30 min at 4 °C. Fusion buffer (RPMI 1640, 0.2% BSA, 10 mM HEPES, and 30 mM succinic acid at pH 5.5) was added to cells for 2 min at 37 °C to induce viral FFWO. To measure pH-independent plasma membrane fusion of viral particles, control wells were incubated with RPMI 1640 supplemented with 0.2% BSA and 10 mM HEPES (pH 7.4) for 2 min at 37 °C. The medium was aspirated, and cells were incubated at 37 °C in DMEM supplemented with 5% FBS, 10 mM HEPES, and 20 mM $NH_4Cl$ (pH 7.4). After 16 hr, the medium was removed, the cells were imaged on an CTL ImmunoSpot S6 Analyzer (CTL), and GFP-positive cells were counted in each well.

## HTNV hamster challenge and passive transfer

Female Syrian hamsters (*Mesocricetus auratus*) 6–8 weeks of age (Envigo, Indianapolis, IN) were anesthetized by inhalation of vaporized isoflurane using an IMPAC6 veterinary anesthesia machine. Once anesthetized, hamsters were injected with the indicated concentration of HTNV diluted in 0.2 mL PBS by the i.m. (caudal thigh) route using a 25-gauge, 1-inch needle at a single injection site. Antibodies were administered at the indicated concentrations and days to anesthetized hamsters by i.p. injection using a 23-gauge, 1-inch needle. Vena cava blood draws occurred on anesthetized hamsters within approved blood collection limitations. Terminal blood collection occurred under KAX (ketamine-acepromazine-xylazine) anesthesia and prior to pentobarbital sodium for euthanasia. All animal procedures were conducted in an animal biosafety level (ABSL-3) laboratory.

## HTNV pseudovirion neutralization assay (PsVNA)

The PsVNA using a non-replicating VSVΔG-luciferase pseudovirion system was performed as previously described (*Kwilas et al., 2014*). The plasmid used to produce the HTNV pseudovirions was pWRG/HTN-M(co) (*Hooper et al., 2020*).

## Nucleoprotein (N) ELISA

The ELISA used to detect N-specific antibodies (N-ELISA) was described previously (*Hooper et al., 1999*; *Elgh et al., 1998*).

## Isolation of RNA and real-time PCR

RNA isolation from homogenized lung and kidney tissues and real-time PCR were conducted as previously described (*Perley et al., 2019*).

## In situ hybridization tissue studies

Lung and kidney tissue sections were fixed, stained, and mounted as previously described (*Brocato et al., 2021*). Slides were scored by intensity ranging from 0 to 3, with median values presented (*Figure 7—figure supplement 1*).

## ANDV hamster challenge

Animal challenge studies were conducted in the ABSL-4 facility of the Galveston National Laboratory. The animal protocol for testing of mAbs in hamsters was approved by the Institutional Animal Care and Use Committee (IACUC) of the University of Texas Medical Branch at Galveston (UTMB) (protocol #1912091). 8-week-old female golden Syrian hamsters (*Mesocricetus auratus*) (Envigo) were

inoculated with 200 PFU of Andes hantavirus (strain Chile-9717869) by intramuscular (i.m.) route on day 0. Animals (n=6 per group) were treated with 10 mg/kg of rANDV-34, rANDV-5, or rDENV 2D22 (a negative control dengue virus-specific antibody) by the i.p. route on days 2 and 5 after virus inoculation. Body weight and body temperature were measured each day, starting at day 0. On day 28 post-challenge, all animals were euthanized with an overdose of anesthetic (isoflurane or ketamine/xylazine) followed by bilateral thoracotomy.

## Quantification and statistical analysis

Statistical analysis for each experiment is described in Materials and methods and/or in the Figure legends. All statistical analysis was done in Prism v7 (GraphPad) or RStudio v1.3.1073.

### Binding and neutralization curves

$EC_{50}$ values for mAb binding were calculated through a log transformation of antibody concentration using four-parameter dose-response nonlinear regression analysis with a bottom constraint value of zero. $IC_{50}$ values for mAb-mediated neutralization were calculated through a log transformation of antibody concentration using four-parameter dose-response nonlinear regression analysis with a bottom constraint value of zero and a top constraint value of 100. $EC_{50}$ and $IC_{50}$ values were generated from two to three independent experiments and reported as an average value.

### Testing of protective efficacy in Syrian hamsters

Studies were done with 6–8 hamsters per antibody treatment group. Survival curves were created through Kaplan-Meier analysis, and a log-rank (Mantel-Cox) test was used to compare each mAb treatment group to the isotype control (rDENV 2D22) (* $p<0.05$, ** $p<0.01$, *** $p<0.001$, **** $p<0.0001$).

### N-ELISA and PsVNA

Data analyzed using a one-way analysis of variance (ANOVA) with multiple comparisons for experiments with groups of ≤3 groups. In all analyses, a p value of<0.05 was considered statistically significant.

## Resource availability

### Lead contact

Further information and requests for resources and reagents should be directed to and will be fulfilled by the Lead Contact, James E. Crowe, Jr. (james.crowe@vumc.org).

### Materials availability

Materials described in this paper are available for distribution for nonprofit use using templated documents from Association of University Technology Managers 'Toolkit MTAs', available at: https://autm.net/surveys-and-tools/agreements/material-transfer-agreements/mta-toolkit. The reagents used in this study are available by Material Transfer Agreement with Vanderbilt University Medical Centre.

## Acknowledgements

We thank Dr Rachael Wolters, Dr Robert Carnahan, Dr Seth Zost, Dr Naveen Suryadevara, and Dr Pavlo Gilchuk for intellectual contributions and general experimental support, and Ryan Irving, Alex Bunnell, Rachel Nargi and Brandon Somerville for technical, managerial and project management support. The recombinant VSV/SNV and VSV/ANDV reagents were a kind gift from K Chandran and R Jangra. We thank the UTMB Animal Resource Center for the support of the animal study. We thank Mary L Milazzo for facilitating BSL-3 and ABSL-4 training and transferring viral isolates. The work of TBE was supported by NIH training grant 5T32GM008320-30. The work of JWH. and other USAMRIID authors was supported by the Military Infectious Disease Research Program under project number MI210048. Opinions, interpretations, conclusions, and recommendations are those of the author and are not necessarily endorsed by the U.S. Army. EM data collections were conducted at the Center for Structural Biology Cryo-EM Facility at Vanderbilt University. We acknowledge the use of the Glacios cryo-TEM, which was acquired by NIH grant 1 S10 OD030292-01.

# Additional information

## Competing interests

James E Crowe: has served as a consultant for Luna Labs USA, Merck Sharp & Dohme Corporation, Emergent Biosolutions, GlaxoSmithKline and BTG International Inc, is a member of the Scientific Advisory Board of Meissa Vaccines, a former member of the Scientific Advisory Board of Gigagen (Grifols) and is founder of IDBiologics. The laboratory of J.E.C. received unrelated sponsored research agreements from AstraZeneca, Takeda, and IDBiologics during the conduct of the study. The other authors declare that no competing interests exist.

## Funding

| Funder | Grant reference number | Author |
|---|---|---|
| National Institute of Allergy and Infectious Diseases | 5T32GM008320 | Taylor B Engdahl |
| Military Infectious Diseases Program | MI210048 | Jay W Hooper |
| NIH Office of the Director | S10 OD030292 | James E Crowe |

The funders had no role in study design, data collection and interpretation, or the decision to submit the work for publication.

## Author contributions

Taylor B Engdahl, Conceptualization, Data curation, Formal analysis, Investigation, Methodology, Writing – original draft, Writing – review and editing; Elad Binshtein, Rebecca L Brocato, Natalia A Kuzmina, Lucia M Principe, Nathaniel S Chapman, Irene A Zagol-Ikapitte, Joseph X Reidy, Andrew Trivette, Investigation, Methodology, Writing – review and editing; Steven A Kwilas, Robert K Kim, Monique S Porter, Laura S Handal, Summer M Diaz, Investigation, Writing – review and editing; Pablo Guardado-Calvo, Resources, Writing – review and editing; Félix A Rey, Resources; Minh H Tran, Formal analysis, Investigation; W Hayes McDonald, Supervision, Investigation; Jens Meiler, Supervision; Alexander Bukreyev, Resources, Supervision, Methodology, Project administration, Writing – review and editing; Jay W Hooper, Resources, Supervision, Funding acquisition, Investigation, Project administration, Writing – review and editing; James E Crowe, Conceptualization, Resources, Supervision, Funding acquisition, Writing – original draft, Project administration, Writing – review and editing

## Author ORCIDs

Taylor B Engdahl (ID) http://orcid.org/0000-0002-6280-4405
Steven A Kwilas (ID) http://orcid.org/0000-0003-0383-3879
Pablo Guardado-Calvo (ID) http://orcid.org/0000-0001-7292-5270
Félix A Rey (ID) http://orcid.org/0000-0002-9953-7988
Jens Meiler (ID) http://orcid.org/0000-0001-8945-193X
Alexander Bukreyev (ID) http://orcid.org/0000-0002-0342-4824
James E Crowe (ID) http://orcid.org/0000-0002-0049-1079

## Ethics

Animal challenge studies were conducted in the ABSL-4 facility of the Galveston National Laboratory. The animal protocol for testing of mAbs in hamsters was approved by the Institutional Animal Care and Use Committee (IACUC) of the University of Texas Medical Branch at Galveston (UTMB) (protocol #1912091).

## Decision letter and Author response

Decision letter https://doi.org/10.7554/eLife.81743.sa1
Author response https://doi.org/10.7554/eLife.81743.sa2

## Additional files

### Supplementary files
• Supplementary file 1. Summary table of electron microscopy statistics.

• MDAR checklist

### Data availability
Original/source data for all Figures in the paper is available at Mendeley Data at https://doi.org/10.17632/s5d636frww.1.

The following dataset was generated:

| Author(s) | Year | Dataset title | Dataset URL | Database and Identifier |
|---|---|---|---|---|
| Engdahl T | 2023 | Antigenic mapping and functional characterization of New World hantavirus neutralizing antibodies | https://doi.org/10.17632/s5d636frww.1 | Mendeley, 10.17632/s5d636frww.1 |

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
