## [Editor Report]

Antibodies perform a critical function in host defense against viruses and have emerged as major therapeutic tools in modern medicine, as evidenced by the large-scale use of antibody-based therapies during the COVID-19 pandemic. This paper describes the characterization of human antibodies to hantaviruses that have the potential to create devastating epidemics. The results teach us about the viral structures that are targets for neutralization and the results are relevant for vaccine development and antibody therapeutic design. The evidence provided is convincing and the results are important and should be of interest to immunologists, virologists, and those working on antibody engineering and therapeutic antibodies.

---

## [Decision Letter]

**Decision letter after peer review:**

Thank you for submitting your article "Antigenic mapping and functional characterization of human New World hantavirus neutralizing antibodies" for consideration by *eLife*. Your article has been reviewed by 3 peer reviewers, and the evaluation has been overseen by a Reviewing Editor and Arturo Casadevall as the Senior Editor. The reviewers have opted to remain anonymous.

Essential revisions:

All three reviewers were impressed by the quality and importance of your work and felt the paper merited eventual publication in *eLife* after a revision that would require additional experimental data.

1. Additional experimental efforts are requested for the section detailing the mapping of escape variants as well as clarification of methods employed and corrections of inconsistencies in written text.

2. Note the criticisms about the glycosylation data. Reviewers felt that some of the missing data must be included. The data regarding the Gn/Gc epitope needs to be clear since that affects the conclusions of the paper.

3. Additional experimental or written justifications may be required for animal studies given the differences used in the treatment regimens and limited measurements of outcomes (i.e., requested information of viremia, plaque assay, such as titration of the RNA levels in ANDV challenge model in hamsters (Figure 7)).

4. The reviewers were divided on the need to validate your findings with live virus experiments. On one hand, such data would greatly strengthen the findings and your own prior work (ref 18) has shown that significant IC50 titer difference can occur for some mAbs. On the other, we recognize that this would require considerable additional work and that prior data exists that supports the correspondence between systems. If you have such data and can provide it, at least for some of the viruses that would strengthen the paper. One possible compromise is to better highlight the prior results with the authentic virus at the beginning of the Results section so that the reader is aware of what has correlated and not correlated in this system.

I look forward to a revised paper. Please know that I will send it to the reviewers again so I urge you to do all you can to address their comments.

*Reviewer #1 (Recommendations for the authors):*

1. Comments on data processing and figures: The above indicated figures lack the original data from which they have been obtained.

Figure 1b: please indicate the % value obtained from competition and the Pearson correlation used to establish the clusters as indicated in the methods.

Please note that the bar indicated below the figure seems to be inverted in colors (0% competition black, 100% competition white).

Figure 2a: Is there any original data that could be shown to support the data indicated in the table? Definitively, the data shown in the last row of the table should be provided. Please note that the text (lines 193-194) describing escape mutants of ANDV-44 and SNDV-53 (total of 7 mutations) does not coincide with the mutations shown in the table (for each antibody only two mutations are shown).

Also, it is hard for the reader to differentiate between the two methods used to generate the escape mutants. It would important to include the method of the single passage replicates in the methods section. For a better comparison of both methods, the total time used to allow for viral replication in each method may be helpful.

Also, the authors should clarify lines 166-168 "mutations was present … in the viral original preparation´s stock".

Figure 2c: Please provide the original flow cytometry binding data of the antibodies to the different mutant constructs as represented in the table.

Ext. Data Figure 4: Please indicate the peptide coverage achieved in presence and absence of the antibodies and HDX exchange in the linear sequence.

*Reviewer #2 (Recommendations for the authors):*

1. The manuscript requires extensive editing and editing of jargon. Highlights of the issues are as follows:

a. "Orthohantaviruses" is not a word. Proper use of this is when refereeing to the genus Orthohantavirus (in italics) Hence it is not plural. The term "hantaviruses" is the common name and may be used instead. Or one may state, hantaviruses, a member of the Orthohantavirus genus, family Hantaviridae ….. please correct throughout.

b. Sin nombre orthohantavirus – in italics_(SNV) is the species not Sin Nombre. Sin nombre virus is the common name.

c. Andes orthohantavirus (ANDV) is the species not Andes. Andes virus is the common name.

d. Correct for all names that are referred to as species.

e. Spell out new or Old World hantaviruses; there is no need for the acronym as it is used only a few times.

f. Ribavirin is approved.

g. Line 49- not hantavirus species- use Orthohantaviruses. There is the family Hantaviridae and the authors are only focused on this genus.

h. S2- please help the reader- what are S2 cells?

i. PSVNA- please spell out- see sentence 367 for example. Perhaps restate as neutralizing antibodies were measured using an ANDV/VSV pseudovirus. This needs to be explicitly stated so the reader can follow what is being done. See figure legends also and add information.

j. Figure 1a. State what antigen is being used.

k. Figure legends do not stand alone and hence they need to walk the reader through the figure. It is very hard to interpret based on the information given.

l. Figure 5 a and b legend are missing in my version.

m. Figure 5 c legend – what is real-time analysis?

n. See figure legends and edit so the reader can follow what has been done. The descriptions are too brief and informative.

o. Remove jargon in methods- RTCA for example. These are just standard assays and do not need new acronyms; see also FFWO- that is also not needed.

p. Line 320- add publication being referred to in this sentence.

q. The manuscript focuses on SNV and ANDV yet the majority of cases in South America are in Brazil. Brazil is not mentioned. The reader may not appreciate the extensive number of cases in Brazil. Do these antibodies cross-react with viruses from this region. The authors should edit the manuscript accordingly to provide readers a balanced perspective. When one includes cases in Brazil, they may be the same as in Europe each year.

r. This list is not comprehensive- please review manuscript and correct throughout.

2. Experimental weaknesses

a. In the section on mapping escape variants, it is unclear why the focus would be on the VSV pseudotyped viruses. The proof of escape variants should be done with the live virus to avoid artifacts with the hanta/vsv. The VSV is a different shape, has a different replication kinetic, and the glycoproteins on the hantaviruses are not packed on the surface. Hence this approach is not acceptable alone. It must be confirmed with the actual virus.

b. The animal studies are key to evaluation of the therapeutic potential of the antibodies. Why did the authors use real-time qPCR and not evaluate the level of live virus by plaque assay. Please include limit of detection on the graphs or in the figure legends.

c. In the animal studies, the time of treatment differs for each but the explanation of why the treatment times or doses are not provided. This needs to be stated for each study and clarified.

3. Weaknesses in Premise and Conclusions

a. The abstract states that the humoral response to hantaviruses is incomplete understood. As there is an extensive body of work on the Old World hantaviruses, what is being suggested here? Do the authors refer to people or animal models or both?

b. The authors have identified and mapped novel monoclonal antibodies that bind to the glycoproteins of hantaviruses, however the in vitro neutralization conclusions are based on pseudotyped viruses which do not reflect the native virion. The authors need to show the same level of activity using live virus.

*Reviewer #3 (Recommendations for the authors):*

1. The data that show inhibition of viral attachment or fusion is based on VSV pseudostyles. However, as the authors note, there are differences in antibody potency in VSV pseudotypes versus the authentic virus. Therefore, authentic virus assays need to be conducted for Figure 5.

2. The hantavirus Gn/Gc antigens were produced in insect (S2) cells. The glycosylation patterns are very likely to be quite different from the physiologic protein produced in mammalian cells. This needs to be acknowledged and addressed.

3. In the Results section, it is not clear where the mAbs characterized in this paper were originally isolated. Please clarify.

4. The data from Figure 2a are not clearly described in the text.

5. The structural data from Figure 3 for broadly neutralizing SNV mAbs uses Maporal virus glycoproteins, which is not a human pathogenic hantavirus. The rest of the structure data uses Andes virus glycoproteins. It is not mentioned why this was done. Are similar sites bound by the mAbs in pathogenic New World hantaviruses?

6. It would be useful to show whether there is a correlation between the day 0 neutralizing titer and control of viral replication in the in vivo studies (Figure 6b and 6c, and Figure 6f, g, h).

7. Viremia for the Andes virus in vivo study in Figure 7b should be included.

8. There are some statements that are unclear or potentially inaccurate that should be addressed:

– Line 39 needs to be rephrased to clarify that hantavirus infections in Europe and Asia do not typically have up to 40% lethality rates.

– Lines 226-227 states "Both residues are located on Gn domain B and could represent a critical evolutionary strategy to escape species-specific immunity." This is not well-explained.

– Lines 216-217 and 219-220 appear to be contradictory.

– Lines 400-401 state there are not significant reductions in RNA levels in Figure 6g, but the figure itself show statistical significance.

– Line 97, the PUUV studies were conducted in bank voles, not hamsters, in reference 16.

– Line 90-91 states "Also, these antibodies are not human, and it is unclear what sites are accessible to antibodies during a natural human infection." Please clarify why binding sites on Gn/Gc would be accessible in vitro but not in vivo.

– Lines 123-125, what is the evidence that sequence variability of Gn causes it to be immunodominant?

– Lines 166-168, what is the evidence that escape mutations were already present in the VSV-Gn/Gc stock? Was the stock deep-sequenced?

---

## [Author Response]

Essential revisions:All three reviewers were impressed by the quality and importance of your work and felt the paper merited eventual publication in eLife after a revision that would require additional experimental data.1. Additional experimental efforts are requested for the section detailing the mapping of escape variants as well as clarification of methods employed and corrections of inconsistencies in written text.

We have added additional information on escape mutant generation and mapping, clarified all methods questions, and made edits to correct inconsistencies in the text.

2. Note the criticisms about the glycosylation data. Reviewers felt that some of the missing data must be included. The data regarding the Gn/Gc epitope needs to be clear since that affects the conclusions of the paper.

We have addressed this comment from reviewer 3 below. Although glycosylation is important and does vary based on the expression system, there are very few known glycosylation sites on the hantavirus Gn/Gc proteins, and it is unlikely that any known glycan sites impact antibody binding. We did not observe any notable differences between the binding of our antibodies to Gn/Gc proteins expressed on the surface of mammalian cells (Expi293F) or to the recombinant proteins expressed in insect cells. We also did not identify any known glycosylation sites as critical residues for binding, either through escape mutant mapping or through structural studies. Thus, for these specific neutralizing sites, we do not observe that glycosylation plays a significant role in antibody binding. Two glycans, N138 and N350, that may impact antibodies that bind at the Gn/Gc interface (ANDV-44 and SNV-53) exist in the capping loop region, but we did not have high enough resolution EM data to identify the molecular interactions of SNV-53 or ANDV-44 with Gn. Future work should tease this point out further, but such investigations are beyond the scope of the current paper.

3. Additional experimental or written justifications may be required for animal studies given the differences used in the treatment regimens and limited measurements of outcomes (i.e., requested information of viremia, plaque assay, such as titration of the RNA levels in ANDV challenge model in hamsters (Figure 7)).

As was shown by others and in our earlier experiments (DOI: 10.1016/j.celrep.2021.109086), viremia is not typical for ANDV infection, whereas detection of virus in organs (particularly in liver) is more indicative of systemic infection. In this experiment, we isolated ANDV from livers of all succumbed animals but from none of the survivors. Thus, we cannot make comparisons between treatment groups to determine if viremia was reduced in antibody treatment groups. Based on the survival curves and the neutralizing function of the mAbs tested, it is likely that the antibodies reduced virus in relevant organs.

4. The reviewers were divided on the need to validate your findings with live virus experiments. On one hand, such data would greatly strengthen the findings and your own prior work (ref 18) has shown that significant IC50 titer difference can occur for some mAbs. On the other, we recognize that this would require considerable additional work and that prior data exists that supports the correspondence between systems. If you have such data and can provide it, at least for some of the viruses that would strengthen the paper. One possible compromise is to better highlight the prior results with the authentic virus at the beginning of the Results section so that the reader is aware of what has correlated and not correlated in this system.

We agree that VSV systems may have some limitations for studying hantavirus virology and repeating all in vitro neutralization-type assays with authentic virus could be used to confirm the findings. However, given the logistical difficulties of time delays in repeating all of the assays like the ones described in this study in BSL-3 space that are currently under pressure for COVID-19 Omicron virus and monkeypox virus studies, we do not consider the authentic virus assay repeats critical. The VSV system pseudotyped with Gn/Gc proteins does appear by all measures appropriate for studying antibodies that target the Gn/Gc complex. We and others (DOI: 10.1126/scitranslmed.abl5399) note the largest discrepancies between neutralization potency with Gc-targeting mAbs (such as SNV-24). Gn-targeting mAbs (such as ANDV-5, ANDV34, MIB22, JL16) and quaternary mAbs (such as SNV-53 and ANDV-44) seem to present highly correlated findings in both systems.

Therefore, to clarify this matter, we have added the following text in the beginning of the Results section under the header “Low likelihood of viral escape for neutralizing hantavirus antibodies”.

“Previously, we demonstrated that our mAbs show similar neutralization potencies between VSV pseudotyped viruses and authentic viruses^18^. The one class of mAbs that demonstrated a notable discrepancy were Gc-targeting mAbs, such as SNV-24, which has an ~1,000-fold loss in neutralization potency to authentic viruses. Based on the general similarities between the two systems, we thought the surrogate system was appropriate to study escape mutation generation. Thus, for all neutralization assays, we employed pseudotyped VSVs bearing the glycoproteins Gn/Gc from either SNV or ANDV species.”

We include a note on this in the results section under “Neutralization potency is dependent on bivalent interactions.”

“Although it is important to note that we previously demonstrated that SNV-24 has a ~1,000-fold decrease in IC_50_ neutralization values between the VSV/SNV and the authentic SNV; thus, this finding may not be representative of authentic virus neutralization.”

And we also address this possible concern in the discussion section,

“These accessibility considerations may contribute to the observed incomplete neutralization of authentic hantaviruses by SNV-24^18^. SNV-24 does potently neutralize hantavirus glycoprotein pseudotyped VSVs, but this finding might be due to an alternative arrangement of the glycoprotein spikes on the VSV virion that does not correctly mimic the glycoprotein array on authentic viruses.”

I look forward to a revised paper. Please know that I will send it to the reviewers again so I urge you to do all you can to address their comments.Reviewer #1 (Recommendations for the authors):1. Comments on data processing and figures: The above indicated figures lack the original data from which they have been obtained.Figure 1b: please indicate the % value obtained from competition and the Pearson correlation used to establish the clusters as indicated in the methods.

We have edited Figure 1b to include the values.

Please note that the bar indicated below the figure seems to be inverted in colors (0% competition black, 100% competition white).

We have edited the title on the scale bar in Figure 1b to read: % binding.

Figure 2a: Is there any original data that could be shown to support the data indicated in the table? Definitively, the data shown in the last row of the table should be provided.

The data in the last column of the table was generated qualitatively by adding 200 µL of last-passaged virus with wells containing 10 to 25 µg/mL of the indicated antibody to verify escape for the selection antibody and to verify lack of escape to other mAbs recognizing different antigenic sites. The yes/no determination was made based on presence of GFP+ cells. To provide a quantitative assessment, we conducted an additional experiment and tested each of the mAbs for neutralization at a single concentration for all escape mutant viruses (at a fixed IU/mL concentration). We added these data to Extended Data Figure 2.

Please note that the text (lines 193-194) describing escape mutants of ANDV-44 and SNDV-53 (total of 7 mutations) does not coincide with the mutations shown in the table (for each antibody only two mutations are shown).

We identified two mutations for SNV-53 against VSV/ANDV and VSV/SNV, and then two mutations for ANDV-44 against VSV/ANDV and one mutation against VSV/SNV (for a total of seven). We have clarified these points in the text,

“Variant viruses selected by ANDV-44 or SNV-53 contained mutations in both the Gn ectodomain (K86 on the VSV/SNV background and K356, S97, A96 on the VSV/ANDV background) and Gc domain II near the highly conserved fusion loop (K759, P772 on the VSV/SNV background and Y760 on the VSV/ANDV background)”

Also, it is hard for the reader to differentiate between the two methods used to generate the escape mutants. It would important to include the method of the single passage replicates in the methods section. For a better comparison of both methods, the total time used to allow for viral replication in each method may be helpful.

We have included methods for the single passage mutant generation under the section in the Materials and methods titled, “RTCA single passage escape mutant screening”. For single passage (through RTCA method) we monitored for a drop in cellular impedance over 96 hours. We clarified the timing in the serial passaging methods, although most supernatants were passaged about every 72 hours, some viral mutants took longer to propagate and were passaged up to 168 hours after inoculating the culture. These details are outlined in the Methods section.

Also, the authors should clarify lines 166-168 "mutations was present … in the viral original preparation´s stock".

We have clarified the statement, as follows:

“Due to the high number of replicates demonstrating a neutralization escape phenotype and all of the viruses isolated had the K76T mutation, it is likely that this escape mutation was present at a high proportion in the viral preparation’s original stock.”

Figure 2c: Please provide the original flow cytometry binding data of the antibodies to the different mutant constructs as represented in the table.

The gating strategy used to determine the % binding data is shown in Extended Data Figure 2. All of the % binding data can be found in the Mendeley Data Set:

Engdahl, Taylor (2022), “Antigenic mapping and functional characterization of New World hantavirus neutralizing antibodies”, Mendeley Data, V1, doi: 10.17632/s5d636frww.1

Ext. Data Figure 4: Please indicate the peptide coverage achieved in presence and absence of the antibodies and HDX exchange in the linear sequence.

ANDV peptide sequence coverage maps for ANDV-5, SNV-24, ANDV-34, ANDV-44, and SNV-53 Fab bound to the ANDV GnH/Gc glycoprotein. Peptides covered during HDX-MS analysis are indicated by the blue lines.

**Author response image 1. sa2fig1:** 

**Author response image 3. sa2fig3:** 

HDX exchange heat map in linear sequence showing the difference in relative deuterium-uptake of ANDV GnH/Gc glycoprotein in complex with ANDV-5, ANDV-34, ANDV-44, and SNV-24, SNV53 Fab compared to unbound ANDV. The sequence of ANDV is shown in the heatmap.

**Author response image 4. sa2fig4:** 

**Author response image 5. sa2fig5:** 

**Author response image 6. sa2fig6:** 

**Author response image 7. sa2fig7:** 

**Author response image 8. sa2fig8:** 

Reviewer #2 (Recommendations for the authors):1. The manuscript requires extensive editing and editing of jargon. Highlights of the issues are as follows:a. "Orthohantaviruses" is not a word. Proper use of this is when refereeing to the genus Orthohantavirus (in italics) Hence it is not plural. The term "hantaviruses" is the common name and may be used instead. Or one may state, hantaviruses, a member of the Orthohantavirus genus, family Hantaviridae ….. please correct throughout.

We have removed the one instance in which “Orthohantaviruses” was used.

b. Sin nombre orthohantavirus – in italics_(SNV) is the species not Sin Nombre. Sin nombre virus is the common name.

We have removed the species indication.

c. Andes orthohantavirus (ANDV) is the species not Andes. Andes virus is the common name.

We have removed the species indication.

d. Correct for all names that are referred to as species.

We have removed the species indication.

e. Spell out new or Old World hantaviruses; there is no need for the acronym as it is used only a few times.

We have removed the acronym.

f. Ribavirin is approved.

We noted that there are no Federal Drug Administration approved therapeutics.

g. Line 49- not hantavirus species- use Orthohantaviruses. There is the family Hantaviridae and the authors are only focused on this genus.

We have removed this designation.

h. S2- please help the reader- what are S2 cells?

Schneider 2 (S2) cells are derived from *Drosophila melanogaster*. We have noted this in the text.

i. PSVNA- please spell out- see sentence 367 for example. Perhaps restate as neutralizing antibodies were measured using an ANDV/VSV pseudovirus. This needs to be explicitly stated so the reader can follow what is being done. See figure legends also and add information.

We have spelled out the acronym in the text and again in the figure legends. We have also then described the methods in the Materials and methods section under “HTNV pseudovirion neutralization assay (PsVNA).”

j. Figure 1a. State what antigen is being used.

All of the antigens are designated in the title of each graph. We have also spelled out the list in the figure caption:

“Binding potency of mAbs to recombinant hantavirus antigens, ANDV Gn^H^/Gc, ANDV Gn^H^/Gc_H953F, MAPV Gn^H^/Gc, ANDV Gn^H^, ANDV Gn^B^, ANDV Gc, expressed in S2 cells.”

k. Figure legends do not stand alone and hence they need to walk the reader through the figure. It is very hard to interpret based on the information given.

We have referred to the journal guidelines and have ensured that all relevant information (sample size, n, statistical significance) is available within each figure legend.

l. Figure 5 a and b legend are missing in my version.

We have fixed the numbering on Figure 5.

m. Figure 5 c legend – what is real-time analysis?

The real-time cell analysis neutralization assay has been described in the Materials and methods under the section “Real-time cell analysis (RTCA) neutralization assay.”

n. See figure legends and edit so the reader can follow what has been done. The descriptions are too brief and informative.

We have referred to the journal guidelines and have ensured that all relevant information (sample size, n, statistical significance) is available within each figure legend.

o. Remove jargon in methods- RTCA for example. These are just standard assays and do not need new acronyms; see also FFWO- that is also not needed.

RTCA and FFWO are variations on standard neutralization assays, thus it is appropriate to indicate which assay is being done. Since we refer to these multiple times, we included the accepted acronyms for the assays.

p. Line 320- add publication being referred to in this sentence.

We have added the citation.

q. The manuscript focuses on SNV and ANDV yet the majority of cases in South America are in Brazil. Brazil is not mentioned. The reader may not appreciate the extensive number of cases in Brazil. Do these antibodies cross-react with viruses from this region. The authors should edit the manuscript accordingly to provide readers a balanced perspective. When one includes cases in Brazil, they may be the same as in Europe each year.

We did not include detailed descriptions of outbreaks by country, only mentioning the case study on a person-to-person outbreak in Argentina. All of the species circulating in Brazil are of the Andes orthohantavirus species, thus using Andes as a representative species is sufficient. Also, this is not a practical request because species native to Brazil are not regularly isolated and used in laboratories and there are 0 full-length coding sequences in NCBI to use for recombinant protein expression – a non-trivial feat even with broadly studied species, such as Andes.

r. This list is not comprehensive- please review manuscript and correct throughout.

We have reviewed and edited the manuscript throughout.

2. Experimental weaknessesa. In the section on mapping escape variants, it is unclear why the focus would be on the VSV pseudotyped viruses. The proof of escape variants should be done with the live virus to avoid artifacts with the hanta/vsv. The VSV is a different shape, has a different replication kinetic, and the glycoproteins on the hantaviruses are not packed on the surface. Hence this approach is not acceptable alone. It must be confirmed with the actual virus.

We acknowledge that the gold standard would be to use authentic hantaviruses, however, we have previously demonstrated in Engdahl *et al.* (DOI: 10.1016/j.celrep.2021.109086) that for the mAbs included in this study, the neutralization potencies are similar between the surrogate system and authentic viruses. Due to the practical considerations of working under BSL-3/4 conditions with live virus and restricted access currently due to Omicron and monkeypoxviruses, we find that using the VSV system is appropriate to investigate the mechanistic basis of antibody neutralization for this study. Future work could confirm the data we generated here with pseudotyped viruses as correlating with authentic viruses, but all prior evidence suggests they will correspond.

b. The animal studies are key to evaluation of the therapeutic potential of the antibodies. Why did the authors use real-time qPCR and not evaluate the level of live virus by plaque assay. Please include limit of detection on the graphs or in the figure legends.

We have updated Figure 6 to include the limit of detection for the qPCR data. Unlike the Andes model, the nonlethal HTNV model does not have survival as an endpoint. Interestingly, the virus appears to cause viremia and seed endothelial cells throughout the animal, and this virus is not cleared from tissues even in the presence of a robust antibody response to infection, including neutralizing antibodies. To quantify the level of virus in the tissues we used qPCR. We also used ISH to get a more qualitative readout on the level of virus remaining in tissues on Day 28. A plaque assay could be used to determine the level of INFECTIOUS virus in those tissues; however, that information is not critical to support the findings of this manuscript.

c. In the animal studies, the time of treatment differs for each but the explanation of why the treatment times or doses are not provided. This needs to be stated for each study and clarified.

The hamster model of ANDV is extremely robust with minimal variation in disease progression and fast lethal outcome associated with infection. The incubation periods are short with the onset of clinical signs on days 9 to 10 and lethal outcome (or euthanasia for high clinical scores) during the following 1 to 2 days. It is important to provide the treatment and keep a high level of antibodies before the clinical onset of disease. In contrast, infection of hamsters with Hantaan virus (HTNV) results in an asymptomatic, disseminated infection. Nevertheless, in both models it generally understood that treatment with antibodies needs to occur before the viremic phase of the infection.

For the ANDV IM, model viremia is first detected on Day 6, so antibody treatment was started on Day 5 in time to prevent lethal disease (DOI: 10.1128/JVI.00238-07). The HTNV IM model is an infection model, not a lethal disease model. Thus, the treatment must have an impact on endpoints other than survival. One of the endpoints is seroconversion to the infection as measured by an&-N ELISA. Previously we (DOI: 10.3389/fmicb.2020.00832) extended postexposure treatment with the positive control antibody used in this experiment, but the latest timepoint with “clean” protection data was Day 3 post-exposure, so that timepoint was chosen for this initial look at the SNV-53 monoclonal antibody in the HTNV infection model.

3. Weaknesses in Premise and Conclusionsa. The abstract states that the humoral response to hantaviruses is incomplete understood. As there is an extensive body of work on the Old World hantaviruses, what is being suggested here? Do the authors refer to people or animal models or both?

The statement was specifically in reference to the human humoral response. We acknowledge and cite the body of literature related to murine, rabbit, and bank vole studies with Old World hantaviruses, but there are currently only three published studies characterizing antibodies isolated from human survivors of natural infection, and very little information of epitopes and mechanism of action—the gap we intended to fill with this work. We have revised the sentence with the qualifier “*human* humoral immune response”.

b. The authors have identified and mapped novel monoclonal antibodies that bind to the glycoproteins of hantaviruses, however the in vitro neutralization conclusions are based on pseudotyped viruses which do not reflect the native virion. The authors need to show the same level of activity using live virus.

Please see response to essential revisions #4 above.

Reviewer #3 (Recommendations for the authors):1. The data that show inhibition of viral attachment or fusion is based on VSV pseudostyles. However, as the authors note, there are differences in antibody potency in VSV pseudotypes versus the authentic virus. Therefore, authentic virus assays need to be conducted for Figure 5.

We agree that in vitro assays with authentic viruses would be a relevant system, however, for practical reasons we used recombinant VSVs bearing the hantavirus glycoproteins. We acknowledge that there may be differences in the organization of the spikes on the virion surface, replication of the virus, and so on, that may impact our results. However, based on previous experiments in Engdahl *et al.* (DOI: 10.1016/j.celrep.2021.109086) comparing neutralization IC_50_ values between VSVs and authentic viruses, Gc-targeting antibodies were the only mAbs that demonstrated a discrepancy between the two systems (~1,000-fold less potent against authentic viruses). The only Gc targeting mAb we include is SNV-24 and we note this discrepancy several times in the text. Although one could repeat all of this work in the future, and we agree that it is desirable confirmatory data to have, it is also not feasible for us at this moment to repeat all work with authentic virus.

2. The hantavirus Gn/Gc antigens were produced in insect (S2) cells. The glycosylation patterns are very likely to be quite different from the physiologic protein produced in mammalian cells. This needs to be acknowledged and addressed.

We agree that the reviewer raises an important point since glycans on virions might vary widely depending on the cells infected, but these studies are beyond the scope of the current manuscript.

3. In the Results section, it is not clear where the mAbs characterized in this paper were originally isolated. Please clarify.

We have added text to the results to clarify where the mAbs came from:

“We previously isolated a panel of 36 mAbs from human survivors that were previously infected with SNV or ANDV, and demonstrated that the mAbs target eight sites on Gn/Gc, multiple of which were neutralizing. Thus, we picked five mAbs from our previously published panel to represent each site to further determine where neutralizing epitopes exist. We also produced recombinant IgG1 forms of two other previously published human mAbs, MIB22 or JL16 (hereafter, rMIB22 or rJL16), based on each antibody’s published heavy or light chain variable genes and cloned them into a human IgG1^19^.”

4. The data from Figure 2a are not clearly described in the text.

We have edited the text to help clarify:

“To do this, we employed two different methods: (1) a high throughput escape mutant generation assay using real-time cellular analysis based on similar assays previously described 21-23 and (2) serial passaging of virus in increasing concentrations of neutralizing antibodies (Figure 2a). Previously, we demonstrated that our mAbs show similar neutralization potencies between VSV pseudotyped viruses and authentic viruses18. The one class of mAbs that demonstrated a notable discrepancy were Gctargeting mAbs, such as SNV-24, which has ~1,000 fold loss in neutralization potency to authentic viruses. Based on the general similarities between the two systems, we thought the surrogate system was appropriate to study escape mutation generation. Thus, for all neutralization assays, we employed pseudotyped VSVs bearing the glycoproteins Gn/Gc from either SNV or ANDV species. For the RTCA escape mutant generation, we tested each antibody at saturating neutralization conditions and evaluated escape based on delayed cytopathic effect (CPE) in each individual replicate well. Thus, if a replicate well demonstrated a delayed drop in cellular impedance, this indicated the presence of an escape mutant virus and was noted out of the total number of replicate wells to give a percentage.”

5. The structural data from Figure 3 for broadly neutralizing SNV mAbs uses Maporal virus glycoproteins, which is not a human pathogenic hantavirus. The rest of the structure data uses Andes virus glycoproteins. It is not mentioned why this was done. Are similar sites bound by the mAbs in pathogenic New World hantaviruses?

We did nsEM with Maporal virus due to practical considerations. When we prepared grids and examined them under the microscope, the MAPV GnH/Gc purified protein was more heterogenous compared to the ANDV GnH/Gc sample and we thought that this occurrence would give us higher quality data. MAPV and ANDV are highly similar, and the general sites targeted by SNV-53 and SNV-24 should not differ substantially between the two species. We do not have high resolution data to determine what the critical contract residues are or if they are conserved for SNV-24 and SNV-53 between the MAPV and ANDV Gn/Gc protein.

6. It would be useful to show whether there is a correlation between the day 0 neutralizing titer and control of viral replication in the in vivo studies (Figure 6b and 6c, and Figure 6f, g, h).

In a previous paper it was shown that the neutralizing antibody level at the time of HTNV exposure predicts protection in the hamster model (Perley et al. 2020). Specifically, a HTNV PsVNA_50_ titer of 157 pre-exposure gave an estimated probability of protection of 0.9 as measured by prevention of seroconversion as measured by an&-N ELISA. In the current study, with very few exceptions, if the sera was positive for neutralizing activity as measured by PsVNA, then the animal was protected from infection as measured by an&-N ELISA, lung and kidney PCR, and lung and kidney ISH. In every case there was a correlation. This finding was true for both the pre- and post-exposure experiments. The Pearson r values ranged from 0.7315 to -0.9384.

**Author response image 9. sa2fig9:** Assay values for individual animals. Individual animal PsVNA50 (log 10) titers were sorted highest to lowest and plotted. Day 28 ELISA titers (log 10), PCR levels (Log 10), and ISH scores (0-3) were also plotted. (A) Data from pre-exposure experiment where antibody was injected 1 day before exposure to HTNV. Sera were collected just prior to challenge with virus. (B) Data from post-exposure experiment where antibody was injected 3 days after exposure to HTNV and sera was collected one day later (Day 4). ISH = in situ hybridization, PCR = quantitative polymerase chain reaction.

**Author response image 10. sa2fig10:** Relationship between level of neutralizing antibodies in serum and protection for pre-exposure experiment. There is a negative correlation for neutralizing antibody levels and infection for all endpoints measured. When antibody is administered 1 day pre-exposure, the level of neutralizing antibody (PsVNA50 log 10) negatively correlates with protection as measured by Day 28 anti-N ELISA, lung PCR, lung ISH, kidney PCR, or kidney ISH. Pearson r values range -0.7315 to -0.8993. An ELISA value <100 (2 log 10), qPCR less than 50 (1.69897 log 10), or an ISH score <1 were considered negative (dotted line). A PsVNA50 titer below the assay limit was given a value of 1.14 (grey shaded area). All hamsters that were negative by PsVNA were infected with HTNV as measured by positive ELISA, Lung and Kidney PCR, and Kidney ISH. There was one hamster that was negative for neutralizing antibodies that was negative by Lung ISH. There were only 3-4 of 23 hamsters positive by PsVNA with positive Lung and or Kidney PCR. Those animals were negative by ISH.

**Author response image 11. sa2fig11:** Relationship between level of neutralizing antibodies in serum and protection for post-exposure experiment. There is a negative correlation for neutralizing antibody levels and infection for all endpoints measured. When antibody is administered 3 days post-exposure, the level of neutralizing antibody (PsVNA50 log 10) negatively correlates with protection as measured by Day 28 anti-N ELISA, lung PCR, lung ISH, kidney PCR, or kidney ISH. Pearson r values range -0.8533 to -0.9384. An ELISA value < 100 (2 log 10), qPCR less than 50 (1.69897 log 10), or an ISH score <1 were considered negative (dotted line). A PsVNA50 titer below the assay limit was given a value of 1.14 (grey shaded area). All hamsters that were negative by PsVNA were infected with HTNV as measured by positive ELISA, Lung and Kidney PCR, and Lung and Kidney ISH. There were only 2 of 30 hamsters positive by PsVNA with positive PCR in lung and/or kidney.

As was shown by others (PMID: 32209676, PMID: 22705798) and in our earlier experiments (PMID: 33951434), viremia is not typical for ANDV infection, whereas detection of virus in organs (particularly in liver) is more indicative. In this experiment, we isolated ANDV from livers of all succumbed animals but from none of the survivors.

7. Viremia for the Andes virus in vivo study in Figure 7b should be included.8. There are some statements that are unclear or potentially inaccurate that should be addressed:– Line 39 needs to be rephrased to clarify that hantavirus infections in Europe and Asia do not typically have up to 40% lethality rates.

We have revised the sentence.

– Lines 226-227 states "Both residues are located on Gn domain B and could represent a critical evolutionary strategy to escape species-specific immunity." This is not well-explained.

We have added an additional sentence for clarification:

“If amino acid changes in highly immunogenic epitopes on Gn domain B do not impact viral fitness, then we may see genetic changes in these sites over time or in the case of a significant outbreak.”

– Lines 216-217 and 219-220 appear to be contradictory.

We have replaced lines 219-220 with:

“We did identify a few single mutations that impacted the binding of a multiple broadly neutralizing mAbs.”

– Lines 400-401 state there are not significant reductions in RNA levels in Figure 6g, but the figure itself show statistical significance.

We have noted the mistake and changed the sentence:

“There was a statistically significant reduction in the RNA levels in the lungs of all SNV53-treated groups. There was a similar readout in the kidneys, except that the SNV-53 5 mg/kg protection was not statistically significant.”

– Line 97, the PUUV studies were conducted in bank voles, not hamsters, in reference 16.

We have changed the text to indicate that the PUUV study was in bank voles.

– Line 90-91 states "Also, these antibodies are not human, and it is unclear what sites are accessible to antibodies during a natural human infection." Please clarify why binding sites on Gn/Gc would be accessible in vitro but not in vivo.

The organization of Gn/Gc on a virion during natural infection likely differs from the epitopes presented on recombinant protein, VSVs, and virus-infected tissues – which was used to immunize animals to generate all animal derived antibodies. We have edited the sentence to clarify that point:

“Also, these antibodies are not human derived and were elicited upon immunization with recombinant protein, VSVs, or virus infected tissue. It is unclear what sites are accessible to antibodies during a natural infection, and if human antibodies target sites that differ from their rodent counterparts.”

– Lines 123-125, what is the evidence that sequence variability of Gn causes it to be immunodominant?

Gn is likely more immunodominant due to its surface exposure, which then leads to increased sequence variability due to humoral pressure. We have edited the sentence to clarify that point:

“However, mAbs targeting Gn were not described, and, due to its level of surface exposure, the N-terminal domain of Gn likely represents a major site of the neutralizing human antibody response^9,17^. Gn demonstrates a high degree of sequence variability than Gc, likely indicating that it is under more immune pressure^5^.”

– Lines 166-168, what is the evidence that escape mutations were already present in the VSV-Gn/Gc stock? Was the stock deep-sequenced?

The stock was not deep-sequenced. The RTCA escape mutant generation does not require multiple passaging, and since escape was seen in most of the replicate wells it is likely – but not sequence confirmed – that these mutations were present in the original stock at some proportion and did not occur due to replication errors under repeated passaging and antibody selection.